# ONE FORWARD IS ENOUGH FOR NEURAL NETWORK TRAINING VIA LIKELIHOOD RATIO METHOD

**Jinyang Jiang**[1,2*]**, Zeliang Zhang**[3*]**, Chenliang Xu**[3†]**, Zhaofei Yu**[4†]**, Yijie Peng**[1,2†]

[1] Wuhan Institute for Artificial Intelligence, Guanghua School of Management, Peking University
[2] Xiangjiang Laboratory    [3] Department of Computer Science, University of Rochester
[4] Institute for Artificial Intelligence, Peking University

`jinyang.jiang@stu.pku.edu.cn`, `{zeliang.zhang,chenliang.xu}@rochester.edu,`
`{yuzf12,pengyijie}@pku.edu.cn`

## ABSTRACT

While backpropagation (BP) is the mainstream approach for gradient computation in neural network training, its heavy reliance on the chain rule of differentiation constrains the designing flexibility of network architecture and training pipelines. We avoid the recursive computation in BP and develop a unified likelihood ratio (ULR) method for gradient estimation with only one forward propagation. Not only can ULR be extended to train a wide variety of neural network architectures, but the computation flow in BP can also be rearranged by ULR for better device adaptation. Moreover, we propose several variance reduction techniques to further accelerate the training process. Our experiments offer numerical results across diverse aspects, including various neural network training scenarios, computation flow rearrangement, and fine-tuning of pre-trained models. All findings demonstrate that ULR effectively enhances the flexibility of neural network training by permitting localized module training without compromising the global objective and significantly boosts the network robustness.

## 1 INTRODUCTION

Since backpropagation (BP) (Rumelhart et al., 1986) has greatly facilitated the success of artificial intelligence (AI) in various real-world scenarios (Song et al., 2021a;b; Sung et al., 2021), researchers are motivated to connect this gradient computation method in neural network training with human learning behavior (Scellier & Bengio, 2017; Lillicrap et al., 2020). However, there is no evidence that the learning mechanism in biological neurons relies on BP (Hinton, 2022). Pursuing alternatives to BP holds promise for not only advancing our understanding of learning mechanisms but also developing more robust and interpretable AI systems. Moreover, the significant computational cost associated with BP (Gomez et al., 2017; Zhu et al., 2022) also calls for innovations that simplify and expedite the training process without heavy consumption.

There have been continuous efforts to substitute BP in neural network training. For example, the HSIC bottleneck (Ma et al., 2020), feedback alignment (FA) (Nøkland, 2016), and neural tangent kernel (NTK) (Jacot et al., 2018) employ additional modules with constraints to compute the gradients without relying on the chain rule of differentiation, thus avoiding BP. While model-based methods remain biologically implausible and suffer from instability and efficiency, perturbation-based methods, including forward-forward (FF) (Hinton, 2022), evolution strategy (ES) (Salimans et al., 2017), and likelihood ratio (LR) (Peng et al., 2022) method, which only involve the forward computation and have little assumption on the model architecture, offer a promising path for exploring biologically plausible BP alternatives.

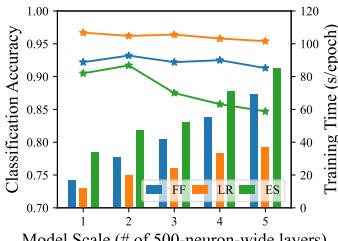

Figure 1: Classification accuracies (curves) and training durations (bars) of three methods till convergence.

---

* These authors contributed equally to this work. Listing order is random.
† Corresponding author.

In the perturbation-based category, the preliminary experiment with multilayer perceptrons (MLPs) (Rumelhart et al., 1986) on the MNIST dataset (LeCun, 1998) indicates that LR might surpass others in terms of efficiency or accuracy, as shown in Fig. 1. FF abandons the classical optimization paradigm to adjust each layer separately, and its incompatibility with other existing techniques hampers efficiency. The weight randomization in ES is more likely to produce infeasible exploration, resulting in instability, and becomes costly when the parameter dimensionality grows. By contrast, training by LR optimizes the model following unbiased gradient estimates without imposing backward recursions and only requires single forward propagation, which can be accelerated by data replication with low consumption and integrated with other training techniques.

We argue that LR is a better alternative to BP for four reasons. First, breaking the reliance on the chain rule, LR requires only one forward evaluation for efficient learning. Second, LR has almost no constraints on the model, including differentiability or traceability, enabling explorations of network architectures. Third, gradient computation among different modules can be independent in LR, which allows for a more flexible training paradigm design. Last, LR can enhance the model robustness by smoothing the loss landscape. However, the application of LR still faces various challenges. Gradient estimation in different networks calls for unified derivation methodology, implementation-friendly conversion, and practical stabilization tricks. Furthermore, the generalization of LR should not be confined just to the level of neural layers. LR has the potential to be employed as a technique at various scales, such as in neuron-wise parallelism or computation graph reorganization.

In our paper, we develop a unified LR (ULR) method on the most general network possible to uncover its essence and application potential. We then generalize pure LR from a special case on MLPs to four network architectures representing different gradient computation challenges. Several variance reduction approaches are first proposed by us in the LR context to significantly stabilize and make model training via only one forward propagation much more practical. Meanwhile, LR is also employed to transform the existing training computation flow of BP, adapting to features for modern devices. We conduct experiments of our ULR with the aforementioned network structures on corresponding tasks and provide two applications, including domain adaption and computation graph rearrangement. Numerical results indicate that ULR effectively enhances the flexibility of neural network training in model and pipeline designing, and significantly improves the model robustness.

## 2 RELATED WORK

Optimization methods for neural network training can be roughly categorized into two types, with and without BP. BP-based optimization methods have been studied for a long time and have been effective in training various neural networks on different tasks. Among these methods, the vanilla BP method is utilized for computing the gradients of the neural network parameters with respect to the global loss function. BP requires the perfect knowledge of computation details between the parameters and loss evaluation to launch a recursive computation, which limits the flexibility of developing neural network architectures and training pipelines. Due to the reliance on the chain rule, this can also be the case for some methods (Baydin et al., 2018) claiming a forward-only training.

Some studies have explored alternative methods without BP for training neural networks by adding extra functional modules to guide model updates. The HSIC bottleneck (Ma et al., 2020) maximizes the Hilbert-Schmidt independence criterion (Wang et al., 2021) to enhance layer independence. NTK (Jacot et al., 2018) introduces intermediary variables to circumvent the limitations of BP. FA (Nøkland, 2016) introduces intermediary variables to overcome BP limitations but faces instability issues. Since these methods change the classical training paradigm or even give up the gradient information, many algorithmic or hardware technologies developed based on BP by predecessors cannot be fully leveraged, making such approaches computationally unfriendly.

Other works still adhere to the original mathematical problem but propose perturbation-based solutions from a stochastic optimization perspective. Spall (1992) proposes a simultaneous perturbation stochastic approximation (SPSA) method, which approximates the gradient by evaluating the objective function twice with perturbations of opposite signs on the parameters. SPSA is typically sensitive to the choice of hyperparameters. Another approach is ES (Salimans et al., 2017), which injects a population of random noises into the model and optimizes the parameters along the optimal direction of noises. Both SPSA and ES suffer from heavy performance drops as the number of parameters increases, which hinders the application to sophisticated models in deep learning.

FF (Hinton, 2022) modifies the input data and works by distinguishing the positive and negative samples for each module but fails to address the weight-sharing and compatibility issue. Unlike a pathwise gradient computed by the BP, Peng et al. (2022) perturb the logit outputs of each neural layer in MLPs and employ the push-out LR technique to estimate the gradient of randomized loss evaluation. However, previous research on LR has primarily focused on MLP training as a whole, which can be recognized as a specific instance of the architecture extension in this paper.

## 3 UNIFIED LIKELIHOOD RATIO METHOD FOR FLEXIBLE TRAINING

### 3.1 GRADIENT ESTIMATION USING PUSH-OUT LR TECHNIQUE

Mainstream neural network training can be roughly divided into a loop with three phases: forward evaluation, gradient computation, and parameter update. LR addresses the second issue as BP does, and it can be integrated with any existing method for the other two phases. We generalize the LR method to the training of generic neural networks and only assume the network to have a modular structure, where each module might consist of single or multiple neural layers. Given a hierarchical neural network, which is sliced into $L$ modules, the input to the $l$-th module is denoted as $x^l \in \mathbb{R}^{d_l}$, $l = 0, \cdots, L-1$, and the output is given by

$$x^{l+1} = \varphi^l(x^l; \theta^l) + z^l, \quad z^l \sim f^l(\cdot),$$

where $x^{l+1}$ is the output of the $l$-th module, $\varphi^l$ and $\theta^l \in \Theta^l$ respectively denote the non-parametric structure and parameter, and $z^l \in \mathbb{R}^{d_{l+1}}$ is a newly added zero-mean noise vector with the density $f^l$. Training the neural network can be regarded as solving such an optimization problem: $\min_{\theta \in \Theta} \sum_{x^0 \in \mathcal{X}} \mathbb{E}\left[\mathcal{L}(x^L)\right]$, where $\mathcal{X}$ denotes the dataset and $\mathcal{L}$ is the loss function. A basic approach is to perform gradient descent following the stochastic gradient estimation of $\mathbb{E}\left[\mathcal{L}(x^L)\right]$ with respect to the parameters. Denote the Jacobian matrix of $y \in \mathbb{R}^{d_y}$ with respect to $x \in \mathbb{R}^{d_x}$ as $D_x y \in \mathbb{R}^{d_y \times d_x}$. Under mild integrability conditions, we can derive the following ULR estimator for the generic network structure:

**Theorem 1.** *Given an input data $x^0$, assume that $g^l(\xi) := f^l(\xi - \varphi^l(x^l; \theta^l))$ is differentiable, and*

$$\mathbb{E}\left[\int_{\mathbb{R}^{d_{l+1}}} \left|\mathbb{E}\left[\mathcal{L}(x^L)|\xi, x^l\right]\right| \sup_{\theta^l \in \Theta^l} \left|\nabla_{\theta^l} g^l(\xi)\right| d\xi\right] < \infty. \tag{1}$$

*Then, we have*

$$\nabla_{\theta^l} \mathbb{E}\left[\mathcal{L}(x^L)\right] = \mathbb{E}\left[-\mathcal{L}(x^L) D_{\theta^l}^\top \varphi^l(x^l; \theta^l) \nabla_z \ln f^l(z^l)\right]. \tag{2}$$

*Proof.* Notice that conditional on $x^l$, $x^{l+1}$ follows a distribution characterized by the density $g^l(\xi)$. With the property of conditional expectation, we can push the parameter $\theta^l$ out of the loss function $\mathcal{L}(x^L)$ and into the conditional density as below:

$$\nabla_{\theta^l} \mathbb{E}_{z^0, \cdots, z^{L-1}}\left[\mathcal{L}(x^L)\right] = \nabla_{\theta^l} \mathbb{E}_{x^l}\left[\int_{\mathbb{R}^{d_{l+1}}} \mathbb{E}_{z^{l+1}, \cdots, z^{L-1}}\left[\mathcal{L}(x^L)|\xi, x^l\right] g^l(\xi) d\xi\right],$$

which is the so-called push-out LR technique. Then, we can obtain

$$\nabla_{\theta^l} \mathbb{E}\left[\mathcal{L}(x^L)\right] = \mathbb{E}\left[\int_{\mathbb{R}^{d_{l+1}}} \mathbb{E}\left[\mathcal{L}(x^L)|\xi, x^l\right] \nabla_{\theta^l} \ln g^l(\xi) g^l(\xi) d\xi\right]$$

$$= \mathbb{E}\left[-\int_{\mathbb{R}^{d_{l+1}}} \mathbb{E}\left[\mathcal{L}(x^L)|\xi, x^l\right] D_{\theta^l}^\top \varphi^l(x^l; \theta^l) \nabla_z \ln f^l(z)|_{z = \xi - \varphi^l(x^l; \theta^l)} g^l(\xi) d\xi\right]$$

$$= \mathbb{E}\left[-\int_{\mathbb{R}^{d_{l+1}}} \mathbb{E}\left[\mathcal{L}(x^L)|\zeta + \varphi^l(x^l; \theta^l), x^l\right] D_{\theta^l}^\top \varphi^l(x^l; \theta^l) \nabla_z \ln f^l(\zeta) f^l(\zeta) d\zeta\right]$$

$$= \mathbb{E}\left[-\mathcal{L}(x^L) D_{\theta^l}^\top \varphi^l(x^l; \theta^l) \nabla_z \ln f^l(z^l)\right].$$

The first equality is justified by the LR technique and the dominated convergence theorem under the given integrability condition (1), the third equality holds by the change of variable, and the rest comes from definitions and the iterative property of the conditional expectation. $\square$

By selecting appropriate noise distributions and excluding the discontinuous or black-box portions in the network from the modules with parameters, we can ensure the differentiability requirements in

Theorem 1. Eq. (2) implies that the gradient estimation for the whole model does not require the recursive backward computation and can be further paralleled with only the final evaluation of $\mathcal{L}(x^L)$ and the local information $x^l$ and $z^l$. As shown in Fig. 2, the application of auto-differentiation is confined within modules, where $D_{\theta^l}\varphi^l(x^l;\theta^l)$ is computed independently from each other with different $l$. ULR also supports an adaptive perturbation by optimizing the parameterized noise distribution $f^l(\cdot;\vartheta^l)$ following the gradient:

$$\nabla_{\vartheta^l}\mathbb{E}\left[\mathcal{L}(x^L)\right] = \mathbb{E}\left[\mathcal{L}(x^L)\nabla_{\vartheta^l}\ln f^l(z^l;\vartheta^l)\right],$$

which could benefit the representation capacity and robustness of the model (Xiao et al., 2022). In the next section, we specify $f^l$ as a multivariate Gaussian distribution with trainable covariance and zero mean for readability, while other distributions, e.g., the exponential family, are also acceptable.

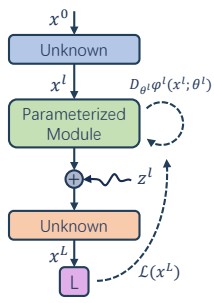

Figure 2: Computation flow with our ULR in generic networks.

## 3.2 Extension to various network architectures

ULR does not impose specific requirements on the modules in the neural network, and gradient computation is localized. Therefore, we can analyze and extend ULR to almost all computational structures from different networks by treating them as single modules. Since these modules can be rewritten as the combination of linear transformations and non-linear processes, we can directly perturb the logit output of the former and estimate the gradients with only one forward propagation, completely avoiding the auto-differentiation technique.

**Convolutional neural networks (CNNs)**: Convolution operation is the representative of the spacial parameter sharing, where the outputs are related to the same kernel. Denote the input as $x = (x^i_{j,k}) \in \mathbb{R}^{c_{in} \times h_{in} \times w_{in}}$, where the superscript represents the channel index. The output is given by

$$v^o_{j,k} = (x * \theta)^o_{j,k} + b^o + z^o_{j,k},$$

where $*$ denotes the convolution operation, $\theta \in \mathbb{R}^{c_{out} \times c_{in} \times h_\theta \times w_\theta}$ and $b = (b^o) \in \mathbb{R}^{c_{out}}$ are the weight and bias terms, $z^o_{j,k} = \sigma^o\varepsilon^o_{j,k}$ is the noise injected to perform ULR, $\sigma^o$ denotes the noise magnitude, and $\varepsilon^o_{j,k}$ is a standard Gaussian random variable. The minimum unit $\theta^{o,i}_{j,k}$ is involved in the computation of the $o$-th output channel. Thus, we push $\theta^{o,i}$ into the conditional density of $v^o$ given $x^i$ and obtain the gradient estimation for each convolutional kernel channel by Theorem 1 as below:

$$\nabla_{\theta^{o,i}}\mathbb{E}\left[\mathcal{L}(v)\right] = \mathbb{E}\left[(\sigma^o)^{-1}\mathcal{L}(v)x^i * \varepsilon^o\right], \tag{3}$$

where the rest of the network and loss function are abbreviated as $\mathcal{L}$. Eq. (3) is quite computationally friendly and can be implemented by the convolution operator in any standard toolkit.

When the kernel tensor is small, the logit perturbation may be costly concerning the parameter size. An alternative way to perturb the neuron is to inject noises into the parameters: $v = x * \hat{\theta} + b$, $\hat{\theta}^{o,i} = \theta^{o,i} + \sigma^o\varepsilon^{o,i}$, where the elements in $\varepsilon^{o,i}$ are independently and normally distributed. Then, we treat $\theta$ as the input and output of an identity mapping module and apply Theorem 1, resulting in

$$\nabla_{\theta^{o,i}}\mathbb{E}\left[\mathcal{L}(v)\right] = \mathbb{E}\left[(\sigma^o)^{-1}\mathcal{L}(v)\varepsilon^{o,i}\right]. \tag{4}$$

Since the variances of Eq. (3) and (4) both depend on the noise dimension, the selection between the two can be determined by the total dimensions of the output and convolutional kernels.

**Recurrent neural networks (RNNs)**: Gradient estimating in RNNs faces the temporal weight-sharing issue. Denote the $t$-th input and hidden states as $x_t \in \mathbb{R}^{d_x}$ and $h_t \in \mathbb{R}^{d_h}$, $t = 1, \cdots, T$. RNN variants can be summarized as a generic structure and the next hidden state is given by

$$h_t = \varphi(u_t, v_t, h_{t-1}), \quad u_t = \theta^{hh}h_{t-1} + b^{hh} + z^{hh}_t, \quad v_t = \theta^{xh}x_t + b^{xh} + z^{xh}_t,$$

where $\theta^{hh}$ and $\theta^{xh}$ are weight matrices, $b^{hh}$ and $b^{xh}$ denote bias vectors, $z^{hh}_t = (\Sigma^{hh})^{\frac{1}{2}}\varepsilon^{hh}_t$, $z^{xh}_t = (\Sigma^{xh})^{\frac{1}{2}}\varepsilon^{xh}_t$, $\varepsilon^{hh}_t$ and $\varepsilon^{xh}_t$ are standard Gaussian random vectors. Taking $\theta^{hh}$ as an example, we first

treat it as different in each time step to apply Theorem 1, and then sum up the step estimates. Denote the rest of the forward process as $\mathcal{L}$ and let $h = (h_1, \cdots, h_T)$. The gradient can be unrolled as

$$\nabla_{\theta^{hh}} \mathbb{E}\left[\mathcal{L}(h)\right] = \mathbb{E}\left[\mathcal{L}(h)(\Sigma^{hh})^{-\frac{1}{2}} \sum_{t=1}^{T} \varepsilon_t^{hh} h_{t-1}^{\top}\right]. \tag{5}$$

Eq. (5) enables the gradients of the RNN variants to be computed in parallel over input sequences without relying on BP Through Time (BPTT), which is one of the main advantage of Transformers (Vaswani et al., 2017) over RNNs. Eq. (5) is derived under a generic structure, and their specific forms could be simplified for different cells. Please refer to Appendix A.3 for details.

**Graph neural networks (GNNs)**: By leveraging the matrix representation of graph structures, gradient estimation towards unstructured inputs can also be derived by ULR. Consider a graph convolutional network (GCN) (Kipf & Welling, 2016) layer with the input $h \in \mathbb{R}^{|\mathcal{V}| \times d_{\text{in}}}$ which is the node feature in graph $\mathcal{G} = (\mathcal{V}, \mathcal{E})$. We can perturb the feature extraction to derive an LR estimator, i.e., $h' = \varphi(Gh\theta + z)$, where $\theta \in \mathbb{R}^{d_{\text{in}} \times d_{\text{out}}}$ is the weight matrix, $G$ is a fixed graph-based matrix for feature aggregation, $\varphi$ is the activation function, and $z$ is the noise matrix. The attention mechanism introduced by graph attention networks (GATs) (Velickovic et al., 2017) is also supported by ULR inside the module. Given a GAT layer with single-head attention, the extracted features of the $i$-th node and the attention coefficient of the $(i, j)$-th edge with injected noises are given by

$$v_i = h_i\omega + \xi_i, \quad u_{i,j} = (v_i, v_j)\,\theta + \zeta_{i,j},$$

where $\omega \in \mathbb{R}^{d_{\text{in}} \times d_{\text{out}}}$ and $\theta \in \mathbb{R}^{2d_{\text{in}}}$ are trainable weights, $\xi_i$ and $\zeta_{i,j}$ are injected noises, and the output features of each node are weighted by normalized attention coefficients among its neighborhood. Since we can isolate the parametric linear transformation as a module, the derivation of ULR gradient estimations in GNNs is analogous to those in previous networks with Theroem 1.

**Spiking neural networks (SNNs)**: Training SNNs suffers from discontinuity in data and activation. Consider an $L$-layer SNN with leaky integrate-and-fire neurons. The time series input of $l$-th layer at time step $t$ is denoted as $x^{t,l} \in \mathbb{R}^{d_l}$. The next membrane potential and spike are given by

$$u^{t+1,l+1} = ku^{t,l+1}(\mathbf{1} - x^{t,l+1}) + \theta^l x^{t+1,l} + z^{t+1,l}, \quad x^{t+1,l+1} = I(u^{t+1,l+1}, V_{\text{th}}),$$

where $\theta^l$ is the synaptic weight, $k$ is the delay factor decided by the membrane time constant, $I$ is the Heaviside neuron activation function with threshold $V_{\text{th}}$, $z^{t,l} = (\Sigma^l)^{\frac{1}{2}} \varepsilon^{t,l}$ with $\varepsilon^{t,l}$ being standard Gaussian random variable. Potentials are integrated into each neuron and will be released as a spike once exceeding the threshold $V_{\text{th}}$, which is a discontinuous process and has to be smoothed when applying BP. However, ULR allows us to exclude this process from the module and apply Theorem 1 to the remaining portion. The spike signal of the last layer at the last time step, namely $x^{T,L}$, is the output of SNN and passed to the loss function $\mathcal{L}$. By handling the sequential weight-sharing in a similar manner as in RNNs, we can obtain $\nabla_{\theta^l} \mathbb{E}[\mathcal{L}(x^{T,L})] = \mathbb{E}[\mathcal{L}(x^{T,L}) \sum_{t=1}^{T} (\Sigma^l)^{-\frac{1}{2}} \varepsilon^{t,l}(x^{t,l})^{\top}]$.

## 3.3 Variance reduction

ULR is a simulation-driven algorithm that relies on the correlation between noise and loss evaluation to estimate the gradient. When the noise dimensionality is huge, this correlation would be too weak to be identified. Hence, we propose several techniques based on the hierarchical structure and Monte Carlo (MC) sampling in neural networks to reduce the estimation variance for efficient training.

**Layer-wise noise injection**: Since ULR does not require recursive gradient computation, it is allowed to independently perturb each layer or even each neuron and estimate the corresponding gradient, which ensures that the correlation is sufficient to produce meaningful estimates. Gradient estimates for modules can be combined to form the gradient for the entire network for a single update, or they can be immediately used for module updates. With the layer-wise perturbation technique, the optimization problem in Section 3.1 is transformed into a series of sub-problems:

$$\min_{\theta^l \in \Theta^l} \mathbb{E}[\mathcal{L}(x^L)|z^{l'} = 0, \,\forall\, l' \neq l], \quad \text{for } l = 0, \cdots, L-1,$$

which can be interpreted as performing stochastic block coordinate descent.

**Antithetic variable**: We employ antithetic sampling to generate the stochastic components in forward propagation, which can effectively reduce the correlation between different evaluations, resulting in the decline of gradient estimation variance. Assume the integrand in Eq. (2) is repeatedly evaluated for $2N$ times. We can generate noises for the first $N$ evaluations, and then flip their signs for the rest. The estimator of Eq. (2) with antithetic variables can be written as

$$\hat{\nabla}_{\theta^l}\mathbb{E}\left[\mathcal{L}(x^L)\right] = -\frac{1}{2N}\sum_{n=1}^{N}D_{\theta^l}^{\top}\varphi^l(x^l;\theta^l)\big(g^l(z^l(n)) + g^l(-z^l(n))\big), \qquad (6)$$

where $g^l(z) = \mathcal{L}(x^L(z))\nabla_z \ln f^l(z)$, and $z^l(n)$ is the noise applied in the $n$-th pair of evaluations.

**Quasi-Monte Carlo (QMC)**: QMC (Soboĺ, 1990) generates noise and evaluates the integrand using low-discrepancy sequences, which can fill the integration region more evenly than the pseudo-random sequence used in standard scientific computing toolkits and produce a better convergence rate. Morokoff & Caflisch (1995) point out that QMC outperforms MC when the integrand is smooth and the dimensionality is relatively small. Given the smoothness of neural networks and the layer-wise perturbation mentioned earlier, it is desirable to try QMC in ULR estimation.

With these variance reduction approaches, the application of ULR becomes practical, and we present the standard version of neural network training through our ULR method as shown in Appendix B.

## 3.4 COMPUTATION GRAPH REARRANGEMENT VIA ULR

BP on high-performance devices usually suffers from deep recursions and huge computation graph scales. Consider a $L$-module deterministic neural network, where the notations are the same as in Section 3.1. The output is given by $x^{l+1} = \varphi^l(x^l; \theta^l)$, and the BP gradient is calculated as below:

$$\nabla_{x^l}\mathcal{L}(x^L) = D_{x^l}^{\top}x^{l+1}\nabla_{x^{l+1}}\mathcal{L}(x^L), \quad \nabla_{\theta^l}\mathcal{L}(x^L) = D_{\theta^l}^{\top}x^{l+1}\nabla_{x^{l+1}}\mathcal{L}(x^L), \qquad (7)$$

for $l = 0, \cdots, L - 1$. Eq. (7) reflects that BP has to be implemented by the auto-differentiation in a recursive manner, as shown in Fig. 3. If we add extra noise to the $L'$-th module, $L' \in [0, L)$, i.e., $x^{L'+1} = \varphi^{L'}(x^{L'}+z; \theta^{L'})$, where $z \sim f(\cdot)$, then we can treat the network as two clusters connected in series and apply Theorem 1 to obtain

$$D_{x^l}^{\top}x^{L'} = D_{x^l}^{\top}x^{l+1}D_{x^{l+1}}^{\top}x^{L'}, \quad \nabla_{\theta^l}\mathbb{E}\left[\mathcal{L}(x^L)\right] = D_{\theta^l}^{\top}x^{l+1}D_{x^{l+1}}^{\top}x^{L'}\mathbb{E}\left[-\mathcal{L}(x^L)\nabla_z \ln f(z)\right], \qquad (8)$$

for $l = 0, \cdots, L' - 1$. Path-wise derivatives of the rest are the same as Eq. (7). As shown in the middle of Fig. 3, Eq. (8) breaks down the single recursion into two parallelizable ones. Analogously, we can introduce noise anywhere in networks to adaptively partition the recursive gradient computation into subsets suitable for parallelization on various devices. More generally, we can split the computation graph by ULR, such as rearranging BPTT computations in RNNs into several parallel mini-BPTT computations to enhance efficiency as shown in the last subfigure of Fig. 3.

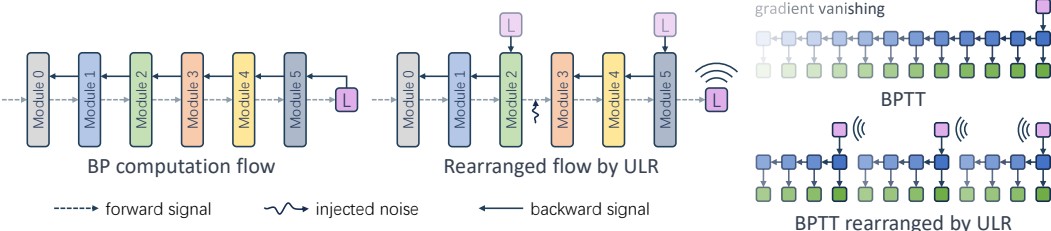

Figure 3: The left subfigure presents the BP computation flow; the middle demonstrates the ULR rearranged computation flow with injected noise; the right showcases the application of ULR for parallelizing gradient computation in RNNs.

## 3.5 RELATIONSHIP BETWEEN ULR AND OTHER METHODS

**ULR and other perturbation-based methods**: Consider a neural module with noise injected into its parameter, i.e., $v = \varphi(x; \theta + z)$, where $x$ and $v$ are the input and output, respectively, $\varphi$ is a non-linear operator, and the noise $z$ follows a density $f$ defined on $\Omega$. Other modules in the forward

computation are abbreviated as a loss function $\mathcal{L}$. Let $\Omega' := \{y : y - \theta \in \Omega\}$. We can obtain the ULR estimator by pushing $\theta$ into the density as below:

$$\nabla_\theta \mathbb{E}\left[\mathcal{L}(v)\right] = \nabla_\theta \int_{\Omega'} \mathbb{E}\left[\mathcal{L}(v)|\xi\right] f(\xi - \theta)d\xi = \mathbb{E}\left[-\mathcal{L}(v)\frac{\nabla_z f(z)}{f(z)}\right],$$

which unifies several previous works (Wierstra et al., 2014; Sehnke et al., 2010; Nesterov & Spokoiny, 2017; Salimans et al., 2017) under the framework of push-out LR method.

**ULR and reinforcement learning**: We first show that the general framework of reinforcement learning (RL) is coherent with the push-out LR method. The RL agent is modeled as a parameterized probability distribution $\pi_\theta$, and $p$ represents the transition kernel of the environment. Denote the action and state at the $t$-th time step as $a_t$ and $s_t$, where $a_t \sim \pi_\theta(\cdot|s_t)$, $s_{t+1} \sim p(\cdot|s_t, a_t)$. The reward is given by $R(s, a)$, which is a function of the state and action sequence. Policy-based RL aims at solving the optimization: $\max_\theta \mathbb{E}\left[R(s, a)\right]$. We can treat $\theta$ as different at each step and push it into the conditional density of $a_t$ given $s_t$ to perform the LR technique, i.e.,

$$\nabla_\theta \mathbb{E}\left[R(s, a)\right] = \sum_{t=0}^{T-1} \nabla_\theta \mathbb{E}\left[\int_{\mathcal{A}} \mathbb{E}\left[R(s, a)|\xi, s_t\right] \pi_\theta(\xi|s_t)d\xi\right] = \mathbb{E}\left[R(s, a) \sum_{t=0}^{T-1} \nabla_\theta \ln \pi_\theta(a_t|s_t)\right],$$

which results in the well-known policy gradient theorem (Williams, 1992). We can derive the RL gradient using the push-out LR and vice versa. Consider the LR-based RNN training discussed earlier, where the input $x_t$ and hidden state $h_{t-1}$ can be viewed as the RL state, and the results of the linear transformation $u_t$ and $v_t$ can be viewed as the actions. The parameterized part and the rest in the RNN cell correspond to the RL agent and the simulation environment, respectively. And Eq. (5) can be obtained directly from the policy gradient theorem. Therefore, RL and deep learning can be unified from the perspective of the push-out LR framework. RL essentially leverages the advantages of LR to overcome the issue of a black-box simulation environment that affects gradient backward propagation. The detailed derivation in this section is presented in Appendix A.

## 4 EVALUATIONS

### 4.1 VERIFICATIONS ON THE SCALABILITY ACROSS VARIOUS ARCHITECTURES

**Experimental settings**: Our study focuses on classification tasks with different modalities using various neural network architectures. For image classification, we experiment with CNNs (Le-Cun et al., 1998) (ResNet-5 and VGG-8) on the CIFAR-10 dataset (Krizhevsky et al., 2009) and SNNs (Ghosh-Dastidar & Adeli, 2009) on the MNIST (LeCun, 1998) and Fashion-MNIST (Xiao et al., 2017) datasets. We use RNN family (RNN (Medsker & Jain, 2001), GRU (Dey & Salem, 2017), and LSTM (Hochreiter & Schmidhuber, 1997)) to classify the articles on the Ag-News dataset (Zhang & Wallace, 2015) and use GNNs (GCN and GAT) to classify the graph nodes on the Cora dataset (Sen et al., 2008). While other non-BP methods, including HSIC, FA, and FF, lack the scalability to all architectures above, we compare our method with two baselines (BP and ES) by using ULR with noise injection only on logits (ULR-L) and weight/logits (ULR-WL) in a hybrid manner. Additionally, results on larger datasets, as well as a comparison in CNN training with omitted non-BP methods, including HSIC, FA, and FF, are presented in Appendices C.6 and C.7.

**Evaluation metrics**: We evaluate training methods by the task performance and robustness in different training contexts. The task performance is characterized as classification accuracies on original datasets. The robustness can be assessed through accuracies on corrupted noisy data generated by multiple kinds of adversarial attacks or natural noise injection.

**Results**: As depicted in Fig. 4, while ES fails to optimize several models (ResNet-5 and GAT), ULR substantially achieves a comparable performance to BP on clean samples, with only a minor gap of $0.22\%$ on average, which presents an efficient gradient estimation performance and optimization stability. Besides, ULR training surprisingly brings a robustness improvement of $9.53\%$ on average compared with BP. Detailed experiment settings and results are in Appendix C.

### 4.2 EVALUATIONS ON THE FLEXIBILITY OF ULR TRAINING

Benefiting from the independence of the gradient estimation inside each module, we can use ULR to rearrange the computation graph, thus achieving lower training consumption. We present two

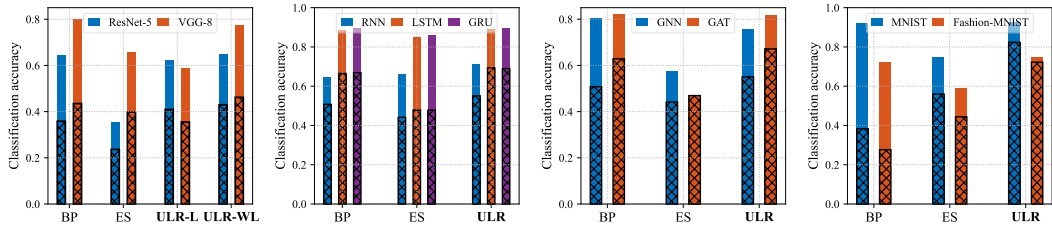

Figure 4: Classification accuracies of trained models in different contexts on benign and corrupted noisy data, represented by blank bars (□) and filled bars (⊠), respectively.

Table 1: Classification accuracy (%) for the domain adaptation task for AlexNet on the Office-31 dataset. We denote $s \rightarrow t$ as adapting the model from the source dataset $s$ to the target dataset $t$, A as the Amazon domain, D as the DSLR domain, and W as the Webcam domain. We take the average training time of different models for efficiency evaluations.

| Method | A→D | A→W | D→A | D→W | W→A | W→D | Average time |
|--------|------|------|------|------|------|------|--------------|
| BP | 51.8 | 58.4 | 65.6 | 94.8 | 68.8 | 88.8 | 202 s |
| ULR | **53.1** | **60.2** | **66.9** | **95.2** | **70.3** | **89.4** | **187** s |

applications, including the training for domain adaption and rearrangement of BPTT in RNNs, to show the training flexibility introduced by our ULR method.

**Demo 1: Domain adaptation.** Multiple domain adaptation methods are proposed to mitigate the domain shifts between the source and target datasets, among which partial domain adaption works on the scene that transfers the partial relevant knowledge from a large-scale source dataset to a small-scale unlabeled dataset. Li et al. (2020) propose the Deep Residual Correction Network (DRCN), which plugs the residual correction block into the source network to mitigate the domain discrepancy between different domains. As shown in Fig. 5, training with BP has

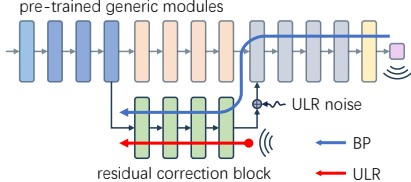

Figure 5: Computation flows in domain adaption with BP and ULR.

to traverse the pre-trained model till the inserted part, while ULR can skip the calculation in late layers and directly compute the gradients of inserted blocks for optimization. As shown in Tab. 1, we use DRCN with both methods to adapt a pre-trained AlexNet on three domains of the Office-31 dataset (Saenko et al., 2010). We compare the accuracies on different domains and time consumption to demonstrate the advances of our method. ULR achieves better performance in terms of both the classification accuracy (1.2% ↑) and training efficiency (1.1× speedups) compared to BP.

**Demo 2: BPTT rearrangement.** RNN training is constrained by the recursive gradient computation of BPTT. Deep recursive computation implies that the parallelism advantages of advanced hardware cannot be utilized, and it can also lead to numerical issues like gradient vanishing. As mentioned in Section 3.4, we perturb the forward propagation of RNNs and decouple the computation graph into several subgraphs using ULR. Although the introduced randomness in training might require extra evaluations to mitigate uncertainty, we achieve considerable efficiency gains at a reasonable level of decoupling due to the parallelism of BPTT on all subgraphs. We test decoupling via ULR on the Ag-News dataset. While vanilla BPTT expends 45 steps on average to compute gradients, ULR partitions the original graph into subgraphs with 5-25 steps. As shown in Fig. 6, ULR boosts BPTT in terms of both the accuracy (up to 4.9% ↑) and efficiency (up to 3.7× speedups), especially when cutting the graph into 15-step subgraphs.

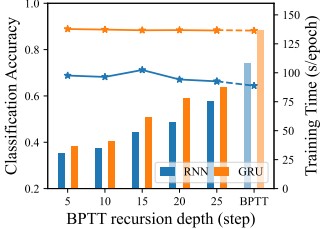

Figure 6: Classification accuracies (curves) and training durations (bars) of BP and ULR.

### 4.3 ABLATION STUDY

We study the effect of perturbing different parts of neurons and variance reduction techniques. In our experiments, we use MC sampling to generate the noise for experiments except when we claim

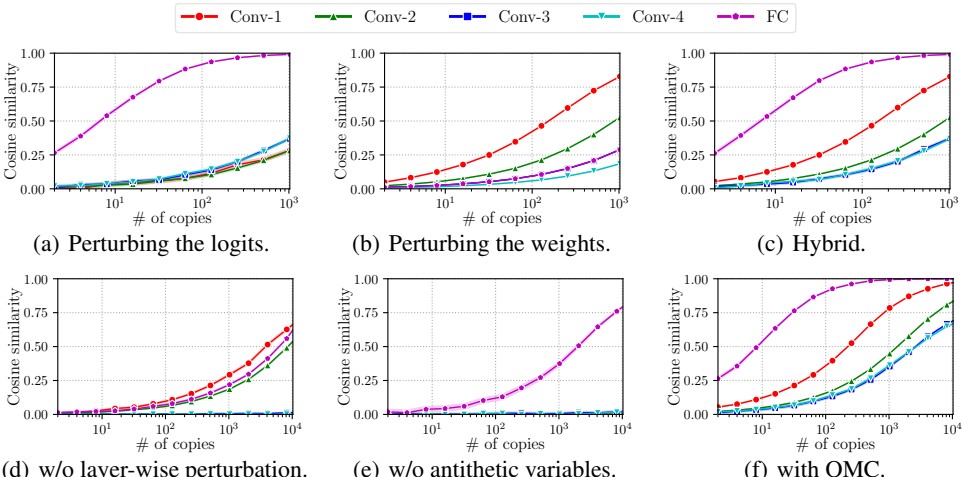

Figure 7: Ablation study on the effect of the perturbation on different parts of neuron networks and the variance reduction techniques. Ablation study on more aspects can be found in Appendix C.10.

otherwise. We conduct experiments using ResNet-5 on the CIFAR-10 dataset for the ablation study. We use our ULR method to estimate the gradient and compare it with that calculated by BP. We compute the cosine similarity to evaluate the performance. The average cosine similarity can serve as a measure of estimation variance. A similarity closer to 1 indicates a lower variance.

**Impact of the perturbation on weights and logits**: We present the gradient estimation performance when perturbing the logits only in Fig. 7(a), the weights only in Fig. 7(b), weights and logits in a hybrid manner in Fig. 7(c). In comparison between Fig. 7(a) and Fig. 7(b), it is observed that perturbing the logits results in a higher cosine similarity between the gradients calculated by ULR and BP for the last three layers, while perturbing the weights yields better gradient estimation with the same number of copies for the first two layers. Being motivated by this phenomenon, we integrate these two perturbing strategies, where we inject the noise on weights for the first two layers and logits for others. As shown in Fig. 7(c), the performance of gradient estimation for all five layers has a significant improvement compared to purely adding perturbation on weights or logits.

**Impact of the variance-reduction techniques**: In Fig. 7(d)-(f), we study the effect of three proposed variance-reduction techniques. Without the layer-wise noise injection, it is difficult to identify the correlation between the noise in each layer and the loss evaluation. As shown in Fig. 7(d), it requires a much larger number of copies to achieve comparable performance on the first two and last layers, and fails to efficiently estimate the gradients of the rest layers. From Fig. 7(e), it can be seen that without the antithetic variable, there are low similarities between the estimated gradients by ULR and that by BP for all convolutional layers even with a large number of copies. In Fig. 7(f), we apply QMC to generate the noise for gradient estimation. Although the results are close to MC, the neuron-level parallelism is allowed by ULR, resulting in a lower dimension of MC integration, which implies that QMC may lead to a more significant improvement.

## 5 CONCLUSION

In our work, we generalize LR to the most universal training paradigm, enabling the extension to a broader range of neural network architectures and exploration of various training pipeline designs. ULR breaks the necessity of the chain rule and only requires forward evaluation during training, thus eliminating the dependency on recursive gradient computations. With variance reduction techniques, our method achieves comparable performance to BP on clean data and exhibits improved robustness. Moreover, we also discuss the relationship between ULR and other perturbation-based methods and unify RL and deep learning from a bidirectional perspective of the general LR framework.

ACKNOWLEDGMENTS

This work was supported in part by the National Natural Science Foundation of China (NSFC) under Grants 72325007, 72250065, and 72022001, as well as the Special Funds for Guiding Local Scientific and Technological Development by the Central Government under Grant 2023EGA035. Z. Zhang and C. Xu were supported by NSF under Grant 2202124 and the Center of Excellence in Data Science, an Empire State Development-designated Center of Excellence. Z. Yu was supported by the Beijing Nova Program under Grant 20230484362. The content of the information does not necessarily reflect the position of the Government, and no official endorsement should be inferred.

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

# A    THEORETICAL DETAILS

## A.1    GRADIENT ESTIMATION FOR PARAMETERIZED NOISE DISTRIBUTIONS

Notice that $z^l$ follows a parameterized noise distribution $f^l(\cdot; \vartheta^l)$, where $\vartheta^l \in \tilde{\Theta}^l$. With the property of conditional expectation, we have

$$\nabla_{\vartheta^l} \mathbb{E}\left[\mathcal{L}(x^L)\right] = \nabla_{\vartheta^l} \int_{\mathbb{R}^{d_{l+1}}} \mathbb{E}\left[\mathcal{L}(x^L)|\zeta\right] f^l(\zeta; \vartheta^l) d\zeta,$$

Under mild integrability conditions, e.g.,

$$\int_{\mathbb{R}^{d_{l+1}}} \left|\mathbb{E}\left[\mathcal{L}(x^L)|\zeta\right]\right| \sup_{\vartheta^l \in \tilde{\Theta}^l} \left|\nabla_{\vartheta^l} f^l(\zeta; \vartheta^l)\right| d\zeta < \infty,$$

we can obtain

$$\begin{aligned}
\nabla_{\vartheta^l} \mathbb{E}\left[\mathcal{L}(x^L)\right] &= \int_{\mathbb{R}^{d_{l+1}}} \mathbb{E}\left[\mathcal{L}(x^L)|\zeta\right] \nabla_{\vartheta^l} \ln f^l(\zeta; \vartheta^l) f^l(\zeta; \vartheta^l) d\zeta \\
&= \mathbb{E}\left[\mathcal{L}(x^L) \nabla_{\vartheta^l} \ln f^l(z^l; \vartheta^l)\right].
\end{aligned} \tag{9}$$

The first equality is justified by the LR technique and the dominated convergence theorem, the second equality comes from definitions and the iterative property of the conditional expectation.

## A.2    GRADIENT ESTIMATION FOR CNNS

The conditional density of $v^o$ given $x^i$ is $f^o(\xi) = \prod_{j=0}^{h_{\text{out}}-1} \prod_{k=0}^{w_{\text{out}}-1} \frac{1}{\sigma^o} \phi\left(\frac{\xi_{j,k} - \tilde{v}_{j,k}^o}{\sigma^o}\right)$. With a specialized formulation of integrability condition (1), i.e.,

$$\mathbb{E}\left[\int_{\mathbb{R}^{h_{\text{out}} \times w_{\text{out}}}} \left|\mathbb{E}\left[\mathcal{L}(v)|\xi, x^i\right]\right| \sup_{\theta_{j,k}^{o,i} \in \mathbb{R}} \left|\frac{\partial}{\partial \theta_{j,k}^{o,i}} f^o(\xi)\right| d\xi\right] < \infty,$$

we can obtain

$$\begin{aligned}
\frac{\partial \mathbb{E}\left[\mathcal{L}(v)\right]}{\partial \theta_{j,k}^{o,i}} &= \frac{\partial}{\partial \theta_{j,k}^{o,i}} \mathbb{E}\left[\int_{\mathbb{R}^{h_{\text{out}} \times w_{\text{out}}}} \mathbb{E}\left[\mathcal{L}(v)|\xi, x^i\right] f^o(\xi) d\xi\right] \\
&= \mathbb{E}\left[\int_{\mathbb{R}^{h_{\text{out}} \times w_{\text{out}}}} \mathbb{E}\left[\mathcal{L}(v)|\xi, x^i\right] \sum_{s=0}^{h_{\text{out}}-1} \sum_{t=0}^{w_{\text{out}}-1} \frac{\partial}{\partial \theta_{j,k}^{o,i}} \ln\left(\frac{1}{\sigma^o} \phi\left(\frac{\xi_{j,k} - \tilde{v}_{j,k}^o}{\sigma^o}\right)\right) f^o(\xi) d\xi\right] \\
&= \mathbb{E}\left[\int_{\mathbb{R}^{h_{\text{out}} \times w_{\text{out}}}} \mathbb{E}\left[\mathcal{L}(v)|\xi, x^i\right] \sum_{s=0}^{h_{\text{out}}-1} \sum_{t=0}^{w_{\text{out}}-1} \frac{x_{j+s,k+t}^i}{\sigma^o} \frac{\xi_{s,t} - \tilde{v}_{s,t}^o}{\sigma^o} f^o(\xi) d\xi\right] \\
&= \mathbb{E}\left[\int_{\mathbb{R}^{h_{\text{out}} \times w_{\text{out}}}} \mathbb{E}\left[\mathcal{L}(v)|\zeta, x^i\right] \sum_{s=0}^{h_{\text{out}}-1} \sum_{t=0}^{w_{\text{out}}-1} \frac{x_{j+s,k+t}^i}{\sigma^o} \zeta_{s,t} \phi(\zeta) d\zeta\right] \\
&= \mathbb{E}\left[\mathcal{L}(v) \sum_{s=0}^{h_{\text{out}}-1} \sum_{t=0}^{w_{\text{out}}-1} \frac{x_{j+s,k+t}^i}{\sigma^o} \varepsilon_{s,t}^o\right].
\end{aligned} \tag{10}$$

The second equality is justified using the LR technique and the dominated convergence theorem. The fourth equality comes from the change of variables, and the last one holds because of the iterative property of the conditional expectation. Eq. (10) provides neuron-wise gradient estimation for the convolutional layer. After rearranging, we can obtain a channel-wise representation as below:

$$\nabla_{\theta^{o,i}} \mathbb{E}\left[\mathcal{L}(v)\right] = \mathbb{E}\left[\frac{1}{\sigma^o} \mathcal{L}(v) x^i * \varepsilon^o\right]. \tag{11}$$

Using Eq. (11), we can estimate the gradient of the convolutional layer at once by utilizing the group convolution functionality available in any machine learning toolkit. The gradient estimation for $b^o$ is

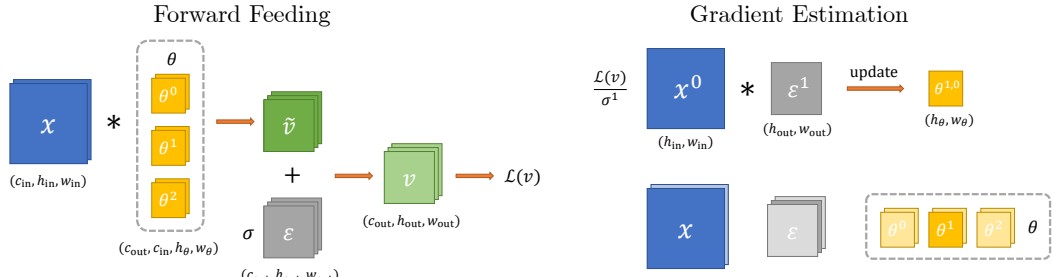

Figure 8: Forward feeding and gradient estimation in convolutional modules.

a corollary of Eq. (10), and we can derive the gradient estimation for noise magnitudes in a similar manner as Eq. (9), i.e.,

$$\frac{\partial \mathbb{E}\left[\mathcal{L}(v)\right]}{\partial b^o} = \mathbb{E}\left[\frac{1}{\sigma^o}\mathcal{L}(v)\sum_{s=0}^{h_{\text{out}}-1}\sum_{t=0}^{w_{\text{out}}-1}\varepsilon_{s,t}^o\right],$$

$$\frac{\partial \mathbb{E}\left[\mathcal{L}(v)\right]}{\partial \sigma^o} = \mathbb{E}\left[\frac{1}{\sigma^o}\mathcal{L}(v)\sum_{s=0}^{h_{\text{out}}-1}\sum_{t=0}^{w_{\text{out}}-1}\left((\varepsilon_{s,t}^o)^2 - 1\right)\right].$$

The forward propagation and gradient estimation by ULR in CNNs are shown in Fig. 8.

### A.3 GRADIENT ESTIMATION FOR RNNS

Since some RNN variants, e.g., LSTMs, have an extra data flow going through $\varphi$ without any parameter, we modify the computation formulation of generic RNNs in Section 3.2 as below:

$$(h_t, c_t) = \varphi(u_t, v_t, h_{t-1}, c_{t-1}), \quad u_t = \theta^{hh}h_{t-1} + b^{hh} + z_t^{hh}, \quad v_t = \theta^{xh}x_t + b^{xh} + z_t^{xh},$$

where $c_t$ denotes another hidden variable. The training objective becomes $\frac{1}{|\mathcal{X}|}\sum_{x\in\mathcal{X}}\mathbb{E}\left[\mathcal{L}(h,c)\right]$, where $h = (h_1, \cdots, h_T)$ and $c = (c_1, \cdots, c_T)$.

Although $\theta^{hh}$ participates in the computation of each $u_t$, the elements in $u = (u_1, \cdots, u_T)$ are autocorrelated so that it is hard to utilize their joint conditional density. Thus, we treat $\theta^{hh}$ in each calculation step as distinct variables, and then sum up the corresponding gradient estimates. Notice that the conditional distribution of $u_t$ given $h_{t-1}$ is $\mathcal{N}(\tilde{u}_t, \Sigma^{hh})$ with the density $f_t(\xi)$, where $\tilde{u}_t = \theta^{hh}h_{t-1} + b^{hh}$. We can unroll the forward pass to perform the push-out LR method as below:

$$\nabla_{\theta^{hh}}\mathbb{E}\left[\mathcal{L}(h,c)\right] = \sum_{t=1}^{T}\nabla_{\theta^{hh}}\mathbb{E}\left[\int_{\mathbb{R}^{d_h}}\mathbb{E}\left[\mathcal{L}(h,c)|\xi, h_{t-1}\right]f_t(\xi)d\xi\right]$$

$$= \sum_{t=1}^{T}\mathbb{E}\left[\int_{\mathbb{R}^{d_h}}\mathbb{E}\left[\mathcal{L}(h,c)|\xi, h_{t-1}\right]\nabla_{\theta^{hh}}\ln f_t(\xi)f_t(\xi)d\xi\right]$$

$$= \sum_{t=1}^{T}\mathbb{E}\left[\int_{\mathbb{R}^{d_h}}\mathbb{E}\left[\mathcal{L}(h,c)|\xi, h_{t-1}\right](\Sigma^{hh})^{-1}(\xi - \tilde{u}_t)h_{t-1}^{\top}f_t(\xi)d\xi\right]$$

$$= \sum_{t=1}^{T}\mathbb{E}\left[\int_{\mathbb{R}^{d_h}}\mathbb{E}\left[\mathcal{L}(h,c)|\zeta, h_{t-1}\right](\Sigma^{hh})^{-\frac{1}{2}}\zeta h_{t-1}^{\top}\phi(\zeta)d\zeta\right]$$

$$= \mathbb{E}\left[\mathcal{L}(h,c)(\Sigma^{hh})^{-\frac{1}{2}}\sum_{t=1}^{T}\varepsilon_t^{hh}h_{t-1}^{\top}\right]. \tag{12}$$

The second equality can be justified by the dominated convergence theorem under a similar integrability condition as Eq. (1), i.e., for $t = 1, \cdots, T$,

$$\mathbb{E}\left[\int_{\mathbb{R}^{d_h}}\left|\mathbb{E}\left[\mathcal{L}(h,c)|\xi, h_{t-1}\right]\right|\sup_{\theta^{hh}\in\mathbb{R}^{d_h\times d_h}}\left|\nabla_{\theta^{hh}}f_t(\xi)\right|d\xi\right] < \infty.$$

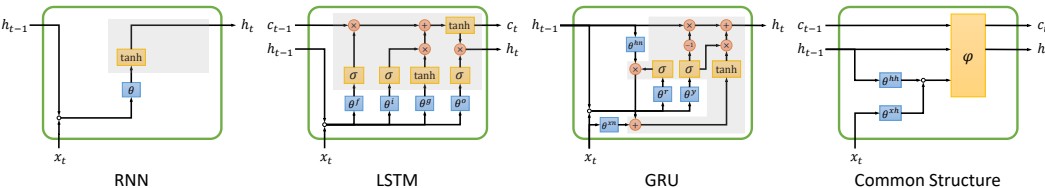

Figure 9: Different RNN modules and their common structure.

By applying the change of variables and the iterative property of the conditional expectation, we can obtain the last equality. The gradient estimates of other parameters in the generic RNN structure can be derived in a similar manner to Eq. (12), i.e.,

$$\nabla_{\theta^{xh}} \mathbb{E}\left[\mathcal{L}(h,c)\right] = \mathbb{E}\left[\mathcal{L}(h,c)(\Sigma^{xh})^{-\frac{1}{2}} \sum_{t=1}^{T} \varepsilon_t^{xh} x_t^{\top}\right],$$

$$\nabla_{b^{hh}} \mathbb{E}\left[\mathcal{L}(h,c)\right] = \mathbb{E}\left[\mathcal{L}(h,c)(\Sigma^{hh})^{-\frac{1}{2}} \sum_{t=1}^{T} \varepsilon_t^{hh}\right],$$

$$\nabla_{b^{xh}} \mathbb{E}\left[\mathcal{L}(h,c)\right] = \mathbb{E}\left[\mathcal{L}(h,c)(\Sigma^{xh})^{-\frac{1}{2}} \sum_{t=1}^{T} \varepsilon_t^{xh}\right],$$

$$\nabla_{\Sigma^{hh}} \mathbb{E}\left[\mathcal{L}(h,c)\right] = \frac{1}{2} \mathbb{E}\left[\mathcal{L}(h,c) \sum_{t=1}^{T} \left((\Sigma^{hh})^{-\frac{1}{2}} \varepsilon_t^{hh} (\varepsilon_t^{hh})^{\top} (\Sigma^{hh})^{-\frac{1}{2}} - (\Sigma^{hh})^{-1}\right)\right],$$

$$\nabla_{\Sigma^{xh}} \mathbb{E}\left[\mathcal{L}(h,c)\right] = \frac{1}{2} \mathbb{E}\left[\mathcal{L}(h,c) \sum_{t=1}^{T} \left((\Sigma^{xh})^{-\frac{1}{2}} \varepsilon_t^{xh} (\varepsilon_t^{xh})^{\top} (\Sigma^{xh})^{-\frac{1}{2}} - (\Sigma^{xh})^{-1}\right)\right].$$

As shown in Fig. 9, the three most widely used RNN cells can be unified as the aforementioned common structure, where hidden states and inputs are delivered to non-linear activation function $\varphi$ after a parametric linear transformation. The modules in the shadows of the first three subfigures of Fig. 9 are specific forms of $\varphi$ in different RNN structures. Since sometimes we can push both the weights of hidden states and inputs into a shared conditional density, we can simplify the perturbation and gradient estimation in different RNN cells.

In vanilla RNN cells, the forward process with perturbation is given by

$$h_t = \tanh\left(\theta^{hh} h_{t-1} + \theta^{xh} x_t + b^{hh} + b^{xh} + z_t\right),$$

where $z_t = (\Sigma)^{\frac{1}{2}} \varepsilon_t$ is shared among the linear transformations of the hidden states and inputs, and $\varepsilon_t^{xh}$ is independent standard Gaussian random vector. The gradient estimates of parameters can be reduced as follows:

$$\nabla_{\theta^{hh}} \mathbb{E}\left[\mathcal{L}(h)\right] = \mathbb{E}\left[\mathcal{L}(h)\Sigma^{-\frac{1}{2}} \sum_{t=1}^{T} \varepsilon_t h_{t-1}^{\top}\right],$$

$$\nabla_{\theta^{xh}} \mathbb{E}\left[\mathcal{L}(h)\right] = \mathbb{E}\left[\mathcal{L}(h)\Sigma^{-\frac{1}{2}} \sum_{t=1}^{T} \varepsilon_t x_t^{\top}\right],$$

$$\nabla_{b^{hh}} \mathbb{E}\left[\mathcal{L}(h)\right] = \nabla_{b^{xh}} \mathbb{E}\left[\mathcal{L}(h)\right] = \mathbb{E}\left[\mathcal{L}(h)\Sigma^{-\frac{1}{2}} \sum_{t=1}^{T} \varepsilon_t\right],$$

$$\nabla_{\Sigma^{hh}} \mathbb{E}\left[\mathcal{L}(h)\right] = \frac{1}{2} \mathbb{E}\left[\mathcal{L}(h) \sum_{t=1}^{T} \left(\Sigma^{-\frac{1}{2}} \varepsilon_t \varepsilon_t^{\top} \Sigma^{-\frac{1}{2}} - \Sigma^{-1}\right)\right].$$

In LSTM cells, the forward propagation with noises is written as

$$f_t = \text{sigmoid}\left(\theta^{xf}x_t + \theta^{hf}h_{t-1} + b^{hf} + b^{xf} + z_t^f\right),$$
$$i_t = \text{sigmoid}\left(\theta^{xi}x_t + \theta^{hi}h_{t-1} + b^{hi} + b^{xi} + z_t^i\right),$$
$$g_t = \tanh\left(\theta^{xg}x_t + \theta^{hg}h_{t-1} + b^{hg} + b^{xg} + z_t^g\right),$$
$$o_t = \text{sigmoid}\left(\theta^{xo}x_t + \theta^{ho}h_{t-1} + b^{ho} + b^{xo} + z_t^o\right),$$
$$c_t = f_t \odot c_{t-1} + i_t \odot g_t, \quad h_t = o_t \odot \tanh\left(c_t\right),$$

where $\theta^{hf}, \theta^{hi}, \theta^{hg}, \theta^{ho} \in \mathbb{R}^{d_h \times d_h}$ and $\theta^{xf}, \theta^{xi}, \theta^{xg}, \theta^{xo} \in \mathbb{R}^{d_h \times d_x}$ are weight matrices in different data flows, $b^{hf}, b^{hi}, b^{hg}, b^{ho}, b^{xf}, b^{xi}, b^{xg}, b^{xo} \in \mathbb{R}^{d_h}$ are bias terms, $z_t^f = (\Sigma^f)^{\frac{1}{2}}\varepsilon_t^f$, $z_t^i = (\Sigma^i)^{\frac{1}{2}}\varepsilon_t^i$, $z_t^g = (\Sigma^g)^{\frac{1}{2}}\varepsilon_t^g$, $z_t^o = (\Sigma^o)^{\frac{1}{2}}\varepsilon_t^o$, $\varepsilon_t^f$, $\varepsilon_t^g$, $\varepsilon_t^i$ and $\varepsilon_t^o$ are independent standard Gaussian random vectors. The gradient estimates of the parameters in the first equality are given by

$$\nabla_{\theta^{hf}}\mathbb{E}\left[\mathcal{L}(h,c)\right] = \mathbb{E}\left[\mathcal{L}(h,c)(\Sigma^f)^{-\frac{1}{2}}\sum_{t=1}^{T}\varepsilon_t^f h_{t-1}^{\top}\right],$$

$$\nabla_{\theta^{xf}}\mathbb{E}\left[\mathcal{L}(h,c)\right] = \mathbb{E}\left[\mathcal{L}(h,c)(\Sigma^f)^{-\frac{1}{2}}\sum_{t=1}^{T}\varepsilon_t^f x_t^{\top}\right],$$

$$\nabla_{b^{hf}}\mathbb{E}\left[\mathcal{L}(h,c)\right] = \nabla_{b^{xf}}\mathbb{E}\left[\mathcal{L}(h,c)\right] = \mathbb{E}\left[\mathcal{L}(h,c)(\Sigma^f)^{-\frac{1}{2}}\sum_{t=1}^{T}\varepsilon_t^f\right],$$

$$\nabla_{\Sigma^f}\mathbb{E}\left[\mathcal{L}(h,c)\right] = \frac{1}{2}\mathbb{E}\left[\mathcal{L}(h,c)\sum_{t=1}^{T}\left((\Sigma^f)^{-\frac{1}{2}}\varepsilon_t^f(\varepsilon_t^f)^{\top}(\Sigma^f)^{-\frac{1}{2}} - (\Sigma^f)^{-1}\right)\right].$$

The gradient estimation for the remaining parameters can be derived analogously.

The forward computation with perturbation in GRU cells is given by

$$r_t = \text{sigmoid}\left(\theta^{xr}x_t + \theta^{hr}h_{t-1} + b^{hr} + b^{xr} + z_t^r\right),$$
$$y_t = \text{sigmoid}\left(\theta^{xy}x_t + \theta^{hy}h_{t-1} + b^{hy} + b^{xy} + z_t^y\right),$$
$$n_t = \tanh\left(\theta^{xn}x_t + b^{xn} + z_t^{xn} + r_t \odot \left(\theta^{hn}h_{t-1} + b^{hn} + z_t^{hn}\right)\right),$$
$$h_t = (1 - y_t)\odot n_t + y_t \odot h_{t-1},$$

where $\theta^{hr}, \theta^{hy}, \theta^{hn} \in \mathbb{R}^{d_h \times d_h}$ and $\theta^{xr}, \theta^{xy}, \theta^{xn} \in \mathbb{R}^{d_h \times d_x}$ are weight matrices in different linear transformations, $b^{hr}, b^{hy}, b^{hn}, b^{xr}, b^{xy}, b^{xn} \in \mathbb{R}^{d_h}$ are bias terms, $z_t^r = (\Sigma^r)^{\frac{1}{2}}\varepsilon_t^r$, $z_t^y = (\Sigma^y)^{\frac{1}{2}}\varepsilon_t^y$, $z_t^{hn} = (\Sigma^{hn})^{\frac{1}{2}}\varepsilon_t^{hn}$, $z_t^{xn} = (\Sigma^{xn})^{\frac{1}{2}}\varepsilon_t^{xn}$, $\varepsilon_t^r$, $\varepsilon_t^y$, $\varepsilon_t^{hn}$ and $\varepsilon_t^{xn}$ are independent standard Gaussian random vectors. The gradient estimates in the first two equalities are the same as in LSTM cells. And the results in the third equality are written as

$$\nabla_{\theta^{hn}}\mathbb{E}\left[\mathcal{L}(h)\right] = \mathbb{E}\left[\mathcal{L}(h)(\Sigma^{hn})^{-\frac{1}{2}}\sum_{t=1}^{T}\varepsilon_t^{hn}h_{t-1}^{\top}\right],$$

$$\nabla_{\theta^{xn}}\mathbb{E}\left[\mathcal{L}(h)\right] = \mathbb{E}\left[\mathcal{L}(h)(\Sigma^{xn})^{-\frac{1}{2}}\sum_{t=1}^{T}\varepsilon_t^{xn}x_t^{\top}\right],$$

$$\nabla_{b^{hn}}\mathbb{E}\left[\mathcal{L}(h)\right] = \mathbb{E}\left[\mathcal{L}(h)(\Sigma^{hn})^{-\frac{1}{2}}\sum_{t=1}^{T}\varepsilon_t^{hn}\right],$$

$$\nabla_{b^{xn}}\mathbb{E}\left[\mathcal{L}(h)\right] = \mathbb{E}\left[\mathcal{L}(h)(\Sigma^{xn})^{-\frac{1}{2}}\sum_{t=1}^{T}\varepsilon_t^{xn}\right],$$

$$\nabla_{\Sigma^{hn}}\mathbb{E}\left[\mathcal{L}(h)\right] = \frac{1}{2}\mathbb{E}\left[\mathcal{L}(h)\sum_{t=1}^{T}\left((\Sigma^{hn})^{-\frac{1}{2}}\varepsilon_t^{hn}(\varepsilon_t^{hn})^{\top}(\Sigma^{hn})^{-\frac{1}{2}} - (\Sigma^{hn})^{-1}\right)\right],$$

$$\nabla_{\Sigma^{xn}}\mathbb{E}\left[\mathcal{L}(h)\right] = \frac{1}{2}\mathbb{E}\left[\mathcal{L}(h)\sum_{t=1}^{T}\left((\Sigma^{xn})^{-\frac{1}{2}}\varepsilon_t^{xn}(\varepsilon_t^{xn})^{\top}(\Sigma^{xn})^{-\frac{1}{2}} - (\Sigma^{xn})^{-1}\right)\right].$$

In simplified schemes, the dimensions of the noise injected into RNN, LSTM and GRU cells are $d_h$, $4d_h$ and $4d_h$, respectively, rather than the original $2d_h$, $8d_h$ and $6d_h$ in generic form.

### A.4 GRADIENT ESTIMATION FOR GNNS

Denote the loss evaluation of GNNs as $\mathcal{L}(h')$, where the parameters in other modules are abbreviated. For GCNs, assume $z_{i,j}$ follows a distribution with the density $f_{i,j}(\xi)$ and let $v = \tilde{v} + z$, where $\tilde{v} = u\theta$ and $u = Gh$. Notice that given $h$, the $j$-th column of $v$ follows a conditional distribution with the density $g_j(\xi) = \prod_{i=0}^{|\mathcal{V}|-1} f_{i,j}(\xi_i - \tilde{v}_{i,j})$. Then, we can derive the gradient estimation by pushing $\theta_{k,j}$ into the conditional density, i.e.,

$$
\begin{aligned}
\frac{\partial}{\partial \theta_{k,j}} \mathbb{E}\left[\mathcal{L}(h')\right] &= \frac{\partial}{\partial \theta_{k,j}} \mathbb{E}\left[\int_{\mathbb{R}^{|\mathcal{V}|}} \mathbb{E}\left[\mathcal{L}(h')|\xi, h\right] g_j(\xi) d\xi\right] \\
&= \mathbb{E}\left[\int_{\mathbb{R}^{|\mathcal{V}|}} \mathbb{E}\left[\mathcal{L}(h')|\xi, h\right] \sum_{i=0}^{|\mathcal{V}|-1} \frac{\partial}{\partial \theta_{k,j}} \ln f_{i,j}(\xi_i - \tilde{v}_{i,j}) g_j(\xi) d\xi\right] \\
&= \mathbb{E}\left[-\int_{\mathbb{R}^{|\mathcal{V}|}} \mathbb{E}\left[\mathcal{L}(h')|\xi, h\right] \sum_{i=0}^{|\mathcal{V}|-1} u_{i,k} \frac{f'_{i,j}(\xi_i - \tilde{v}_{i,j})}{f_{i,j}(\xi_i - \tilde{v}_{i,j})} g_j(\xi) d\xi\right] \\
&= \mathbb{E}\left[-\int_{\mathbb{R}^{|\mathcal{V}|}} \mathbb{E}\left[\mathcal{L}(h')|\zeta, h\right] \sum_{i=0}^{|\mathcal{V}|-1} u_{i,k} \frac{f'_{i,j}(\zeta_i)}{f_{i,j}(\zeta_i)} \prod_{i=0}^{|\mathcal{V}|-1} f_{i,j}(\zeta_i) d\zeta\right] \\
&= \mathbb{E}\left[-\mathcal{L}(h') \sum_{i=0}^{|\mathcal{V}|-1} u_{i,k} \frac{f'_{i,j}(z_{i,j})}{f_{i,j}(z_{i,j})}\right].
\end{aligned}
$$

The second equality is justified using the LR technique and the dominated convergence theorem under a similar integrability condition as Eq. (1), i.e.,

$$
\mathbb{E}\left[\int_{\mathbb{R}^{|\mathcal{V}|}} |\mathbb{E}\left[\mathcal{L}(h')|\xi, h\right]| \sup_{\theta_{k,j} \in \mathbb{R}} \left|\frac{\partial}{\partial \theta_{k,j}} g_j(\xi)\right| d\xi\right] < \infty.
$$

The fourth equality comes from the change of variables, and the last one holds because of the iterative property of the conditional expectation. For GATs, assume $\xi_{i,j}$ and $\zeta_{i,j}$ follow distributions with the densities $f_{i,j}^{\xi}(x)$ and $f_{i,j}^{\zeta}(x)$, respectively, and let $\bar{v}_m = (v_i, v_j)$, where $m = i \times |\mathcal{V}| + j$. The gradient estimates can be derived analogously to that in GCNs, i.e.,

$$
\begin{aligned}
\frac{\partial}{\partial \omega_{k,j}} \mathbb{E}\left[\mathcal{L}(h')\right] &= \mathbb{E}\left[-\mathcal{L}(h') \sum_{i=0}^{|\mathcal{V}|-1} h_{i,k} \frac{f_{i,j}^{\xi}{}'(\xi_{i,j})}{f_{i,j}^{\xi}(\xi_{i,j})}\right], \\
\frac{\partial}{\partial \theta_k} \mathbb{E}\left[\mathcal{L}(h')\right] &= \mathbb{E}\left[-\mathcal{L}(h') \sum_{i=0}^{|\mathcal{V}|-1} \sum_{j=0}^{|\mathcal{V}|-1} \bar{v}_{i \times |\mathcal{V}|+j,k} \frac{f_{i,j}^{\zeta}{}'(\zeta_{i,j})}{f_{i,j}^{\zeta}(\zeta_{i,j})}\right].
\end{aligned}
$$

### A.5 GRADIENT ESTIMATION FOR SNNS

Similar to the derivation for RNNs, we can treat $\theta^l$ in each computation step as different variables, and then sum up the gradient estimates. Let $\tilde{v}^{t,l} = \theta^l x^{t,l}$. Note that conditional on $x^{t,l}$, $v^{t,l} = \tilde{v}^{t,l} + z^{t,l}$ follows a distribution $\mathcal{N}(\tilde{v}^{t,l}, \Sigma^l)$ with the density $f^{t,l}(\xi)$. Then, with a specialization of the integrability condition (1), i.e., for $t = 1, \cdots, T$,

$$
\mathbb{E}\left[\int_{\mathbb{R}^{d_{l+1}}} \left|\mathbb{E}\left[\mathcal{L}(x^{T,L})|\xi, x^{t,l}\right]\right| \sup_{\theta^l \in \mathbb{R}^{d_{l+1} \times d_l}} \left|\nabla_{\theta^l} f^{t,l}(\xi)\right| d\xi\right] < \infty.
$$

we unroll the forward computation and obtain

$$
\begin{aligned}
\nabla_{\theta^l} \mathbb{E}\left[\mathcal{L}(x^{T,L})\right] &= \sum_{t=1}^{T} \nabla_{\theta^l} \mathbb{E}\left[\int_{\mathbb{R}^{d_{l+1}}} \mathbb{E}\left[\mathcal{L}(x^{T,L})|\xi, x^{t,l}\right] f^{t,l}(\xi)d\xi\right] \\
&= \sum_{t=1}^{T} \mathbb{E}\left[\int_{\mathbb{R}^{d_{l+1}}} \mathbb{E}\left[\mathcal{L}(x^{T,L})|\xi, x^{t,l}\right] \nabla_{\theta^l} \ln f^{t,l}(\xi) f^{t,l}(\xi)d\xi\right] \\
&= \sum_{t=1}^{T} \mathbb{E}\left[\int_{\mathbb{R}^{d_{l+1}}} \mathbb{E}\left[\mathcal{L}(x^{T,L})|\xi, x^{t,l}\right] (\Sigma^l)^{-1}(\xi - \tilde{v}^{t,l})(x^{t,l})^{\top} f^{t,l}(\xi)d\xi\right] \\
&= \sum_{t=1}^{T} \mathbb{E}\left[\int_{\mathbb{R}^{d_{l+1}}} \mathbb{E}\left[\mathcal{L}(x^{T,L})|\zeta, x^{t,l}\right] (\Sigma^l)^{-\frac{1}{2}}\zeta(x^{t,l})^{\top}\phi(\zeta)d\zeta\right] \\
&= \mathbb{E}\left[\mathcal{L}(x^{T,L}) \sum_{t=1}^{T} (\Sigma^l)^{-\frac{1}{2}}\varepsilon^{t,l}(x^{t,l})^{\top}\right].
\end{aligned}
$$

And the gradient estimation for $\Sigma^l$ can be given by

$$
\nabla_{\Sigma^l}\mathbb{E}\left[\mathcal{L}(x^{T,L})\right] = \frac{1}{2}\mathbb{E}\left[\mathcal{L}(x^{T,L}) \sum_{t=1}^{T}\left((\Sigma^l)^{-\frac{1}{2}}\varepsilon^{t,l}(\varepsilon^{t,l})^{\top}(\Sigma^l)^{-\frac{1}{2}} - (\Sigma^l)^{-1}\right)\right].
$$

## A.6 GRADIENT ESTIMATION UNDER WEIGHT PERTURBATION

By pushing $\theta$ into the density, we can obtain

$$
\begin{aligned}
\nabla_{\theta}\mathbb{E}\left[\mathcal{L}(v)\right] &= \nabla_{\theta}\int_{\Omega'} \mathbb{E}\left[\mathcal{L}(v)|\xi\right] f(\xi - \theta)d\xi \\
&= \int_{\Omega'} \mathbb{E}\left[\mathcal{L}(v)|\xi\right] \nabla_{\theta} \ln f(\xi - \theta) f(\xi - \theta)d\xi \\
&= -\int_{\Omega'} \mathbb{E}\left[\mathcal{L}(v)|\xi\right] \left.\frac{\nabla_z f(z)}{f(z)}\right|_{z=\xi-\theta} f(\xi - \theta)d\xi \\
&= -\int_{\Omega} \mathbb{E}\left[\mathcal{L}(v)|\zeta\right] \left.\frac{\nabla_z f(z)}{f(z)}\right|_{z=\zeta} f(\zeta)d\zeta \\
&= \mathbb{E}\left[-\mathcal{L}(v)\frac{\nabla_z f(z)}{f(z)}\right],
\end{aligned}
$$

where $\Omega' := \{y : y - \theta \in \Omega\}$, the second equality holds by applying the LR technique and the dominated convergence theorem under the assumption that

$$
\int_{\Omega'} |\mathbb{E}\left[\mathcal{L}(v)|\xi\right]| \sup_{\theta} |\nabla_{\theta} f(\xi - \theta)| \, d\xi < \infty,
$$

and the rest comes from the change of variables and the iterative property of the conditional expectation.

## A.7 RELATIONSHIP BETWEEN ULR AND RL

We have discussed how to derive the key result in policy-based RL by the push-out LR method in our paper. We next provide an inverse perspective on the relationship between LR and RL. Let $\tilde{u}_t = \theta^{hh}h_{t-1} + b^{hh}$ and $\tilde{v}_t = \theta^{xh}x_t + b^{xh}$. The generic form of RNNs can be rewritten as

$$
a_t = \theta o_t + \Sigma^{-\frac{1}{2}}\varepsilon_t, \quad s_{t+1} = \varphi\left(a_t, s_t\right), \tag{13}
$$

where $a_t = (\tilde{u}_t^{\top}, \tilde{v}_t^{\top})^{\top}$, $s_t = (c_{t-1}^{\top}, h_{t-1}^{\top}, x_t^{\top})^{\top}$ and $o_t = (h_{t-1}^{\top}, x_t^{\top})^{\top}$ serve as the action, state and observation in RL context, respectively. The generic RNN cell can be regarded as a partially observable RL scenario. The RL agent is modeled as a single neural layer parameterized by $\theta$ and

$\Sigma$, which is a parametric Gaussian distribution widely used in classical RL with a continuous action space, while the transition kernel of the RL environment is determined by $\varphi$. The cost/reward signal is given by the loss function, which could be episodic feedback like the classification loss, or per-step feedback, such as the mean squared error (MSE) loss for a generation problem. If the reward is not decomposable, e.g., the classification loss, we can obtain the intermediate result in Section A.3 by the vanilla policy gradient theorem, i.e.,

$$\nabla_\theta \mathbb{E}\left[\mathcal{L}(h,c)\right] = \mathbb{E}\left[\mathcal{L}(h,c)\sum_{t=1}^{T}\nabla_\theta \ln f_t(a_t)\right]$$

$$= \sum_{t=1}^{T}\mathbb{E}\left[\int_{\mathbb{R}^{2d_h}}\mathbb{E}\left[\mathcal{L}(h,c)|\xi,o_t\right]\nabla_\theta \ln f_t(\xi)f_t(\xi)d\xi\right],$$

where $f_t(\xi) = f(\xi; o_t)$ is the density of conditional Gaussian distribution $\mathcal{N}\left(\theta o_t^\top, \Sigma\right)$. Otherwise, we have $\mathcal{L}(h,c) = \sum_{t=1}^{T} l(h_t, c_t)$ and can utilize the "reward-to-go" technique in the policy gradient theorem to obtain a more refined result, i.e.,

$$\nabla_\theta \mathbb{E}\left[\mathcal{L}(h,c)\right] = \mathbb{E}\left[\sum_{t=1}^{T}\sum_{t'=t}^{T} l(h_{t'}, c_{t'})\nabla_\theta \ln f_t(a_t)\right]$$

$$= \mathbb{E}\left[\sum_{t=1}^{T}\sum_{t'=1}^{t} l(h_t, c_t)\nabla_\theta \ln f_{t'}(a_{t'})\right]$$

$$= \sum_{t=1}^{T}\sum_{t'=1}^{t}\mathbb{E}\left[\int_{\mathbb{R}^{2d_h}}\mathbb{E}\left[l(h_t, c_t)|\xi,o_t\right]\nabla_\theta \ln f_{t'}(\xi)f_{t'}(\xi)d\xi\right],$$

where the last equality is a direct intermediate result by repeatedly using the push-out LR method.

## B  PSEUDOCODE

---

**Algorithm 1** Neural Network Training via Unified Likelihood Ratio Method

---

**Input:** Network structure $\{\varphi^l(\cdot; \theta^l)\}_{l=0}^{L-1}$, loss function $\mathcal{L}(\cdot)$, input $x^0$, noise density $\{f^l(\cdot)\}_{l=0}^{L-1}$.
  1: Initialize network parameter $\{\theta^l\}_{l=0}^{L-1}$.
  2: **repeat**
  3:   **for** $l = 0$ **to** $L-1$ **parallelly do**
  4:     Generate noise $z^l \sim f^l(\cdot)$ with QMC optional.
  5:     Compute $x_+^{L,l}$ and $x_-^{L,l}$ only with $z^l$ and $-z^l$ injected to the $l$-th layer, respectively.
  6:     Update $\theta^l$ by estimated gradient following Eq. (6) with loss values $\mathcal{L}(x_+^{L,l})$ and $\mathcal{L}(x_-^{L,l})$.
  7:   **end for**
  8: **until** network parameter converges.
**Output:** Network parameter $\{\theta^l\}_{l=0}^{L-1}$.

---

## C  EXPERIMENTAL DETAILS

### C.1  GENERIC EXPERIMENT SETTINGS

**Platform**: We conduct experiments in a computational platform with PyTorch 1.14.0 and eight NVIDIA RTX A6000 GPUs. Each A6000 GPU has 48 GB of memory.

**Datasets**: In our paper, we focus on the classification task. We conduct experiments of ResNet-5 and VGG-8 on the CIFAR-10 dataset, which consists of $60,000$ $32 \times 32$ color images in 10 classes, with $5,000$ images for training and $1,000$ for testing per class. For experiments on the RNN family, we evaluate the proposed method on the Ag-News dataset with 4 classes, which consists of $30,000$ in news articles for training and $1,900$ for testing in each class. For the GNN family, we

Table 2: Results of ResNet-5 and VGG-8.

| Models | Alg. | Ori. | Natural Noise | | | Adversarial Noise | | | Distribution shift | |
|---|---|---|---|---|---|---|---|---|---|---|
| | | | Gaussian | Uniform | Poisson | FGSM | I-FGSM | MI-FGSM | Grey | RanMask |
| ResNet-5 | BP | 64.4 | 59.4 | 60.7 | 44.2 | 12.1 | 1.1 | 1.7 | 53.5 | 54.1 |
| | ES | 35.1 | 35.1 | 35.0 | 29.5 | 20.8 | 4.1 | 4.9 | 25.3 | 34.3 |
| | ULR-L | 62.3 | 61.4 | 61.9 | 50.9 | 26.0 | 6.4 | 9.4 | 55.9 | 55.5 |
| | ULR-WL | 64.8 | 64.2 | 64.6 | 55.3 | 26.2 | 6.7 | 9.4 | 58.2 | 59.1 |
| VGG-8 | BP | 79.8 | 69.4 | 76.4 | 37.8 | 32.4 | 1.2 | 0.9 | 67.8 | 61.9 |
| | ES | 65.8 | 63.7 | 65.4 | 35.6 | 25.7 | 2.7 | 2.7 | 61.9 | 60.5 |
| | ULR-L | 58.7 | 50.6 | 56.2 | 30.7 | 27.9 | 11.5 | 10.5 | 48.9 | 47.6 |
| | ULR-WL | 77.3 | 72.7 | 76.7 | 42.9 | 34.6 | 2.8 | 3.5 | 71.4 | 64.9 |

consider the Cora dataset with 7 classes, which consists of 140 nodes for training and $1,000$ nodes for testing. For SNN, we conduct experiments on the MNIST and Fasion-MNIST datasets, which consist of $60,000$ $28 \times 28$ grey images for training and $10,000$ images for testing. We conduct the domain adaption experiments on the Office-31 dataset, which consists of 3 domains with image size $224 \times 224$, including the Amazon, DSLR, and Webcam.

## C.2 EXPERIMENTS FOR CNNS

**ResNet-5**: The ResNet-5 has 5 layers, including 4 convolutional layers and 1 fully connected layer. The residual connection is between the third and fourth convolutional layers. For the convolutional layers, we set the number of kernels as 8, 16, 32, and 32, respectively, all the kernel sizes as $3 \times 3$ with the stride as 1, and the activation function as ReLU. We use the batch size as 100. For the fully connected layers, we set the number of neurons as 10 for classification with 10 classes. For all the compared training methods, we use the Adam optimizer with the initialized learning rate as $1 \times 10^{-3}$, and train the models with 200 epochs. For the ES and ULR training, we set the $\sigma$ for each layer initialized as $1 \times 10^{-3}$, $1 \times 10^{-3}$, $1 \times 10^{-1}$, $1 \times 10^{-1}$, $1 \times 10^{-1}$, respectively. Due to the high gradient estimation variance, we set the number of copies as $1,000$ for all layers in the ES training method for sufficient optimization. We set the number of copies for each layer as 100, 100, 200, 200, and 50 in ULR training methods.

**VGG-8**: The VGG-8 has 8 layers, including 6 convolutional layers and 2 fully connected layers. For the convolutional layers, we set the number of kernels as 16, 16, 32, 32, 64, and 64, respectively, all the kernel sizes as $3 \times 3$ with the stride as 1, and the activation function as ReLU. We use the batch size as 100. For the fully connected layers, we set the numbers of neurons as 256 and 10, respectively. For all the compared training methods, we use the Adam optimizer with the initialized learning rate as $1 \times 10^{-3}$ and train the models with 200 epochs. For the ES and ULR training, we set the $\sigma$ for each layer initialized as $1 \times 10^{-3}$, $1 \times 10^{-3}$, $1 \times 10^{-3}$, $1 \times 10^{-3}$, $1 \times 10^{-2}$, $1 \times 10^{-2}$, $1 \times 10^{-2}$, and $1 \times 10^{-2}$, respectively. Due to the high gradient estimation variance, we set the number of copies as $1,500$ for all layers in the ES training method for sufficient optimization. We set the number of copies for each layer as 100, 200, 400, 800, 400, 200, 100, and 50 in ULR training methods.

**Evaluation criteria**: We evaluate all the methods for CNNs on the following performance criteria: 1) Task performance: the classification accuracy on the benign samples (Ori.); 2) Natural noise robustness: the classification accuracy on the natural noise corrupted dataset, where we adopt the Gaussian, uniform, and Poisson noises; 3) Adversarial robustness: the classification accuracy on the adversarial examples, which are crafted by the fast gradient sign method (FGSM), iterative fast gradient sign method (I-FGSM), and momentum-based iterative fast gradient sign method (MI-FGSM); 4) distribution shift robustness: the classification accuracy on the corrupted dataset with different distribution corresponding to the original dataset, where we transform the colored dataset into the grey images and apply the random mask to images respectively for the construction of out-of-distribution (OOD) datasets. For the adversarial attacks, we set the maximum perturbation for each pixel as $8/255$, and the maximum number of iterations as 5 for I-FGSM and MI-FGSM. For all the evaluations, we consider the corruption on the whole testset .

Table 3: Results of the RNN family.

| Attack | RNN | | | GRU | | | LSTM | | |
|---|---|---|---|---|---|---|---|---|---|
| | BP | ES | ULR | BP | ES | ULR | BP | ES | ULR |
| Ori. | 64.4 | 65.9 | 70.9 | 88.2 | 84.6 | 88.9 | 89.4 | 85.7 | 89.4 |
| RanMask | 53.4 | 53.4 | 56.7 | 63.9 | 59.2 | 65.5 | 60.1 | 56.2 | 62.1 |
| Shuffle | 62.7 | 63.2 | 69.0 | 87.1 | 80.2 | 88.3 | 88.9 | 82.6 | 89.3 |
| PWWS | 50.0 | 36.5 | 54.0 | 58.5 | 32.0 | 63.0 | 59.0 | 31.5 | 62.5 |
| GA | 37.0 | 23.5 | 40.5 | 56.5 | 20.0 | 60.5 | 59.5 | 21.0 | 62.0 |

**Results**: As shown in Tab. 2, ULR-WL can achieve comparable performance to BP with an improvement of $0.4\%$ for ResNet-5 and a minor drop of $2.6\%$ for VGG-8. Meanwhile, ULR-WL has a significant improvement in the adversarial robustness of $8.1\%$ on average. In ResNet-5, the classification accuracy relies more on the training of the later layers, where adding noise to logits incurs less cost, thus ULR-L outperforms ES and achieves similar performance as ULR-WL. In VGG-8, the situation is exactly the opposite. And the success of ULR-WL in CNNs comes from combining the strengths of both noise injection modes. Moreover, our proposed variance reduction methods are also compatible with ES. The accuracies of ResNet-5 and VGG-8 trained by ES can be boosted to $46.9\%$ and $67.2\%$, respectively, indicating the effectiveness of our tricks in variance reduction. However, even with these tricks, ES struggles to optimize neural networks effectively, demonstrating a lower capacity in gradient estimation compared to ULR.

## C.3  EXPERIMENTS FOR RNNS

For all the studied models, including the RNN cell, GRU cell, and LSTM cell, we use the Glove vector as the pre-trained embedding with the dimension of $100$. We set the number of hidden units as $64$ and the number of units for classification as $4$. We use the Adadm optimizer with the initialized learning rate as $1 \times 10^{-3}$, and train the models with $100$ epochs. We set the batch size as $10$. For both the ES and ULR training methods, we set the number of copies as $200$.

**Evaluation criteria**: We evaluate all the methods for RNNs on task performance and robustness. For the robustness evaluation, we adopt two corruptions, 1) RanMask, in which we randomly mask a fixed ratio of $90\%$ of the words in the whole article; 2) Shuffle, which random shuffles words in sentences; 3) adversarial attacks, including the probability-weighted word saliency (PWWS) and genetic algorithm (GA). We set the maximum ratio of word substitutions as $25\%$. Due to the low efficiency of adversarial text attacks, we consider $200$ examples for evaluation on adversarial attacks and $1,000$ for other perturbations.

**Results**: The evaluation results of the RNN family can be shown in Tab. 3. Due to the gradient vanishing problem in RNN for long sentences, the conventional BP method can not train the neural network well and it only achieves an accuracy of $64.7\%$. Without relying on the chain-rule computation, the ULR method can mitigate the gradient vanishing problem and improve the performance of RNN significantly by $5.5\%$. For training GRU and LSTM, the ULR method can achieve comparable performance on the clean dataset. On the other hand, neural network training using the ULR method can achieve better robustness compared with the BP, namely $2.3\%$ under the RanMask corruption, $2.63\%$ under the Shuffle corruption, and $4.0\%$ under the PWWS attack.

## C.4  EXPERIMENTS FOR GNNS

For all the studied GNN models, we use the Adam optimizer with the learning rate $1 \times 10^{-2}$. Due to the limited size of dataset, we set the batch size as the data size, i.e., the whole $140$ nodes. We train the models for $50$ epochs.

**GCN**: The GCN network has three layers, with the number of neurons as 1433 (input layer), 32, and 7 (output layer), respectively. For the ULR and ES training methods, we set the $\sigma$ as $1 \times 10^{-1}$ for all layers and the number of copies as $100$.

Table 4: Results of the GNN family.

| Attack | GCN | | | GAT | | |
|--------|-----|-----|-----|-----|-----|-----|
| | BP | ES | ULR | BP | ES | ULR |
| Ori. | 80.4 | 57.3 | 75.6 | 82.2 | 34.4 | 81.8 |
| R.A. | 62.5 | 48.2 | 68.9 | 70.4 | 30.1 | 77.3 |
| Dice | 63.2 | 45.7 | 65.4 | 71.3 | 28.9 | 74.6 |

Table 5: Results of SNN.

| Attack | MNIST | | | Fashion-MNIST | | |
|--------|-------|-----|-----|---------------|-----|-----|
| | BP | ES | ULR | BP | ES | ULR |
| Ori. | 91.8 | 74.6 | 92.3 | 72.3 | 58.6 | 74.6 |
| FGSM | 42.6 | 59.5 | 81.4 | 38.0 | 46.2 | 73.0 |
| I-FGSM | 31.5 | 55.2 | 82.7 | 23.1 | 44.6 | 71.8 |
| MI-FGSM | 40.8 | 53.4 | 83.0 | 21.9 | 42.3 | 72.0 |

**GAT**: The GAT network has 2 layers. The first layer consists of 4 attention heads computing 32 features each (for a total 128 features), followed by an exponential linear unit activation. The second layer has a single attention head that computes 4 features for classification. For both ES and ULR training methods, we set $\sigma$ as $1 \times 10^{-1}$. For the ES training method, we set the number of copies to 100. For the ULR training method, we set the number of copies as 1.

**Evaluation criteria**: We evaluate all the methods for GNNs on task performance and robustness. For the robustness evaluation, we adopt two adversarial attacks, including randomly injecting fake edges into the graph (R.A.) and disconnecting internally or connecting externally (DICE). For both the R.A. and DICE attacks, we perturb the edge ratio from $10\%$ to $100\%$ in $10\%$ increments compared to the original number of edges and report the average results. For all the evaluations, we consider the corruption on the whole test set .

**Results**: As shown in Tab. 4, compared with ES, ULR achieves a comparable performance relative to BP, with a performance gap of $4.8\%$ for GCN and $0.4\%$ for GAT on the clean dataset. ULR further brings an average of $4.3\%$ improvement for GCN and an average of $5.1\%$ improvement for GAT on the model robustness. Moreover, the GAT training by ULR does not require any additional copy of data, which has a similar computational cost compared to BP.

## C.5 EXPERIMENTS FOR SNNS

The SNN has three layers with the number of neurons as $784$, $50$, and $10$. For the spiking computation, we set the size of the time window as $5$, the decay factor as $0.5$, the threshold as $0.3$, and the length as $0.5$. We use the Adadm optimizer with the initialized learning rate as $1 \times 10^{-3}$, and train the models with 200 epochs. We use the batch size as 100. For both the ES and ULR training methods, we set the number of copies as 200.

**Evaluation criteria**: We evaluate the performance of different optimization methods by task performance and adversarial robustness. We use FGSM, I-FGSM, and MI-FGSM to craft the adversarial examples on MNIST and Fashion-MNSIT datasets, where the maximum perturbation for each pixel is 0.1 for all the attack methods, the max number of iterations is 15, and the step size is 0.01 for I-FGSM and MI-FGSM. For all the evaluations, we consider the corruption on the whole test set.

**Results**: As shown in Tab. 5, ES fails in training SNN, and the ULR method can achieve a higher classification accuracy on clean and adversarial datasets. Specifically, ULR has an average improvement of $1.55\%$ on two clean datasets and $44.3\%$ on adversarial examples. The activation function in SNN is discontinuous, which hinders the application of the chain rule-based BP method, and the BP adopted here is based on an approximated computation. The unbiasedness of the ULR gradient estimation leads to superior results.

## C.6 VERIFICATION ON LARGER DATASET

To give a further scalability evaluation of the ULR method, we train the ResNet-9 to classify images in the CIFAR-100 dataset. We select BP, ES, ULR-L, and ULR-WL as the baselines. We take the supported maximum number of data copies for gradient estimation in ES, ULR-L, and ULR-WL. We present the results in Tab. 6. From the numerical results, it can be observed that while ES fails to optimize network parameters on the large-scale CIFAR-100 dataset, ULR achieves a comparable performance to BP, with only a minor gap of $0.3\%$.

Table 6: Classification accuracies of ResNet-9 on the CIFAR-100 dataset.

| Methods | BP | ES | ULR-L | ULR-WL |
|---------|------|------|-------|--------|
| Acc. | 65.2 | 27.3 | 59.2 | 64.9 |

## C.7 COMPARISON WITH OTHER METHODS

To comprehensively compare different optimizations, we further train the ResNet-5 on the CIFAR-10 dataset using BP, FF, FA, HSIC, ES, ULR-L, and ULR-WL. As shown in Fig. 10, approaches based on stochastic gradient estimation, especially the two ULR methods, can achieve comparable performance with BP. ULR-WL surpasses the runner-up method, HSIC, which belongs to another training strategy category, with a significant improvement of $9.0\%$. It should be noticed that FF fails to optimize the model due to the weight-sharing issue, which neither ES nor ULR will suffer from during the training process.

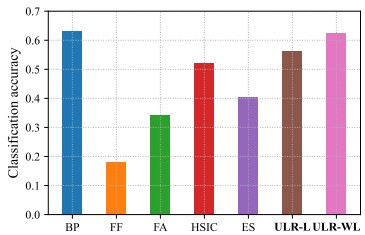

Figure 10: Classification accuracies of ResNet-5 on the CIFAR-10 dataset using different optimization methods.

## C.8 EXPERIMENTS FOR DOMAIN ADAPTION

We use the pre-trained AlexNet in Pytorch for domain adaption experiments and add a residual correction block, consisting of two linear layers with ReLU activation function, after the pooling layer. We use the Adam optimizer with the initialized learning rate $1 \times 10^{-4}$. For all models, we set the training epochs as $50$ and the batch size as $64$.

## C.9 EXPERIMENTS FOR BPTT REARRANGEMENT

We train the RNN and GRU on the Ag-News dataset to verify the performance of BPTT rearrangement via ULR. For all the models, we set the learning rate as $1 \times 10^{-3}$ with Adam optimizer and train the models for $100$ epochs for sufficient convergence. While the average length of texts in the Ag-News dataset is around $45$ words, which corresponds to the average length of the gradient computation graph as $45$ steps, we split the original gradient computation graphs into subgraphs by ULR with depths of 5, 10, 15, 20, and 25, respectively, to give a comprehensive study and comparison on the rearrangement performance.

## C.10 EXTENDED ABLATION STUDY

Following the same setting in our main text, we provide a more extensive ablation study on the VGG-8. The details of the network structure have been introduced in Section C.2. Without specific nomination, we perturb the network without QMC in a hybrid manner for gradient estimation, where we inject the noise on weights for the first four layers and on logits for other layers.

Fig. 11(a)-(c) demonstrate the impact of the perturbation on logits, weights, and the hybrid of weights and logits. It can be noted that compared with adding perturbation on logits and weights, the hybrid manner can achieve a much better gradient estimation accuracy both in the first and last several layers.

Fig. 11(d)-(f) present the impact of the initialization for noise magnitude $\sigma$. Following the design of the aforementioned hybrid noise injection mode, we perform ULR with $\sigma$ set to $1 \times 10^{-1}$, $1 \times 10^{-2}$, and $1 \times 10^{-3}$, respectively. The optimal choice of $\sigma$ varies for different noise injection modes. We can see that $1 \times 10^{-2}$ is more suitable for initializing $\sigma$ for the first several convolutional layers to achieve an acceptable gradient estimation accuracy. In contrast, the last several layers, especially the fully connected layers, are robust to the initialization of $\sigma$, which maintains high gradient estimation accuracy for all selected values.

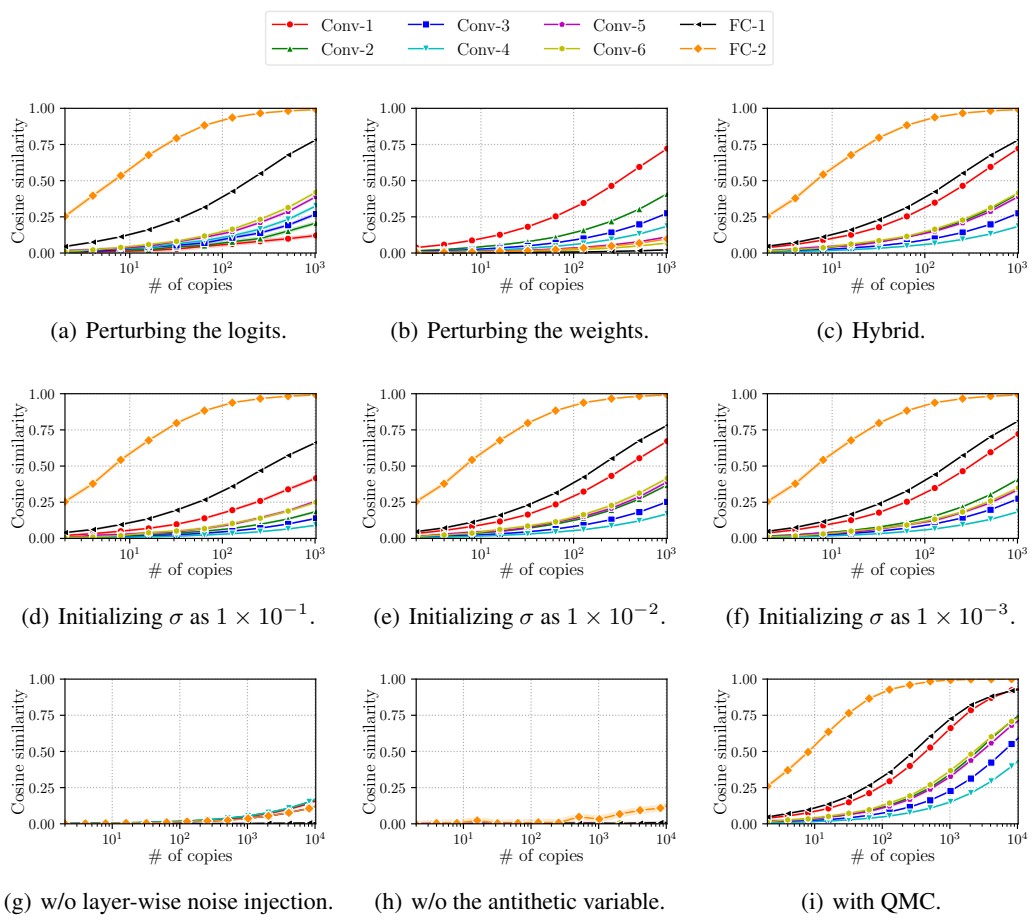

Figure 11: Ablation study on the effect of the perturbation on different parts of neurons, the initialization for noise magnitude $\sigma$, and the variance reduction techniques.

Fig. 11(g)-(i) show the impact of the variance-reduction techniques. From the results in the figures, all the proposed methods, including layer-wise perturbation, antithetic variable, and the use of QMC, play an important role on improving the gradient estimation accuracy.

In a word, the conclusion from the ablation study on VGG-8 is consistent with that on ResNet-5.

Furthermore, we conduct another ablation study on perturbation adapting, where the magnitudes of injected noise are optimized following the gradient estimators derived in Appendices A.2 and A.3. Under the same setting in Section 4.1, we respectively use ULR with and without perturbation adapting to train ResNet-5 and VGG-8 on the CIFAR-10 dataset, as well as RNN, GRU, and LSTM on the Ag-News dataset. As reported in Tab. 7, perturbation adapting boosts the performance of neural networks trained by our ULR method.

Table 7: Classification accuracies with different perturbing strategies in ULR training.

| Dataset | Model | Perturbation | |
|---------|-------|--------------|-------|
| | | Adaptive | Fixed |
| CIFAR-10 | ResNet-5 | 64.8 | 61.3 |
| | VGG-8 | 77.3 | 75.9 |
| Ag-News | RNN | 70.9 | 69.2 |
| | GRU | 88.9 | 86.6 |
| | LSTM | 89.4 | 87.8 |

