# OpenReview forum: "One Forward is Enough for Neural Network Training via Likelihood Ratio Method"
_ICLR.cc/2024/Conference — ICLR 2024 poster_

### Official Review · Reviewer_3HaG · 2023-10-18

**Soundness:** 2 fair
**Presentation:** 2 fair
**Contribution:** 2 fair
**Rating:** 6
**Confidence:** 2

**Summary:**

This paper proposes a unified likelihood ratio (ULR) method that extends the existing likelihood ratio (LR) method to deal with neural network models other than MLPs, including CNNs, RNNs, GNNs, and SNNs. ULR injects noise into the output of each layer, which they call a logit, to estimate the gradient of parameters in the layer instead of using the usual back-propagation (BP), which is theoretically justified by the push-out technique proposed in the LR paper.
Several techniques are proposed to reduce the variance of gradient estimation by noise injection, such as layer-wise injection, antithetic variables, and quasi-Monte Carlo.
Experimental results suggest that ULR allows the training of (relatively shallow) neural networks whose performance is comparable to the model trained with BP.
In addition, models trained with ULR are shown to be more robust to inputs corrupted by adversarial or natural noise.

**Strengths:**

- The existing LR method is extended to deal with various NN architectures, including CNNs, RNNs, GNNs, and SNNs.
- Variance reduction methods are proposed and empirically shown to be effective.
- Experiments are performed on CNNs, RNNs, GNNs, and SNNs where ULR is compared to BP.

**Weaknesses:**

- It is difficult, at least for me, to follow the technical details of the proposed ULR for the following reasons:
    - In Section 3.1, several expectations ($\mathbb{E}$) appear to explain the proposed method. But I could not understand in which variables and distributions these expectations are defined.
    - Conditional expectations such as $\mathbb{E}[\mathcal{L}(x^L)|\xi, x^l]$ are also difficult to understand exactly.
    - What is $J_{\theta^l}$ in Eq.(2), which is not defined in the manuscript? Jacobian of $x^{l+1}$ with respect to $\theta^l$?
- In experiments, ULR is only applied to relatively shallow NNs, such as ResNet-5 or VGG-8.

**Questions:**

- As in the title of the paper, it is claimed that ULR requires only one forward computation to estimate the gradient. However, following Algorithm 1 in the paper, it seems to me that ULR requires additional multiple forward computations with respect to $z^l$, whose size is specified in the appendix as ranging from 50 to 800, to compute the loss values. If this is true, I wonder if ULR really reduces the computational effort compared to BP.

- In section 3.1, it is suggested to optimize the noise distribution parameter. But it seems a bit strange to me to optimize the noise distribution with respect to the loss value. Instead, I think the noise distribution should be optimized to improve the gradient estimation.

- In Figure 4, does ULR-WL for ResNet-5 mean the hybrid model explained in Section 4.3? What about the details of ULR-WL for VGG-8?

- In page 4: $b \in \mathbb{R}^{c_\text{out}}$ -> $b^O \in \mathbb{R}^{c_\text{out}}$

---

> ### Author Response · Authors · 2023-11-19
>
> ### Responses to Weaknesses
>
> > **AW1:** Thank you for your feedback. We will modify the formulas and definitions for readability in the revision. We provide a step-by-step expansion of the loss expectation, explicitly annotating the random variables, to further illustrate details in the derivation of ULR as follows: $$ \mathbb E_{z^0,\cdots, z^{L-1}}[\mathcal L (x^L)]
> = \mathbb E_{x^l}[\mathbb E_{z^l,\cdots, z^{L-1}}[\mathcal L (x^L)|x^l]]
> = \mathbb E_{x^l}[\mathbb E_{x^{l+1}}[\mathbb E_{z^{l+1},\cdots, z^{L-1}}[\mathcal L (x^L)|x^{l+1},x^l]|x^l]]
> = \mathbb E_{x^l}[\int_{\mathbb R ^{d_{l+1}}} \mathbb E_{z^{l+1},\cdots, z^{L-1}}[\mathcal L (x^L)|\xi,x^l ]g^l(\xi)d\xi],$$ where the first two equalities are justified by the law of iterated expectation, and the third equality comes from the definition. Moreover, the notation $J_{\theta^l} \varphi^l(x^l;\theta^l)$ represents the Jacobian matrix of the original output $\varphi^l(x^l;\theta^l)$ with respect to $\theta^l$, but not that of the perturbed result $x^{l+1}$.
>
> > **AW2:** Although ULR has better scalability than previous methods and can support training larger models, **the application of pure ULR on large neural networks might not be the most economical,** as stochastic optimization requires multiple evaluations to improve performance. Thus, in our paper, we conduct experiments with pure ULR as well as other baselines on relatively shallow neural networks to verify the effectiveness of ULR training. However, as presented in Section 3.4, **ULR can rearrange the gradient computation graph in deep learning** and work with BP, where ULR is used to estimate the gradient of the last part in each module, and BP passes the information throughout the module. This approach has much lower pressure on memory than pure ULR, as well as improves the parallelism of gradient computation in large and deep models, thereby enhancing efficiency, as verified by two demos in Section 4.2.

---

> ### Author Response · Authors · 2023-11-19
>
> ### Responses to Questions
>
> >**AQ1:** Firstly, though only one forward is necessary for ULR, we agree that ULR needs multiple forward evaluations with respect to $z^l$ to enhance the gradient estimation. However, it should be noticed that **forward evaluations with different $z^l$ are independent and thus can be easily parallelized.** We can stack multiple copies of the original data batch into **a large batch for just one forward computation**. The computation with a large batch size can fully utilize the capacity of GPUs. Moreover, we apply multiple tricks to improve estimation efficiency, as shown in Alg. 1, which largely alleviates the issue. There is still significant potential to further reduce consumption, such as by reusing evaluation results through the importance sampling technique.
> >Secondly, **ULR can work in conjunction with BP/BPTT, balancing training duration and memory occupation to improve training efficiency.** As presented in Section 3.4 and the second demo in Section 4.2, we can use ULR to **cut the original long gradient computation chain into multiple independent short graphs**, which do not require a lot of forward computation but can greatly improve parallelism.
> >Thirdly, **ULR saves backward computations on those parts that do not need optimization,** thus reducing computational effort compared to BP. When the network embeds pre-trained models or other, even black-box components, ULR can bypass these parts for gradient calculation, as shown in the first demo in Section 4.2.
>
> > **AQ2:** The training process with perturbation-based methods essentially aims at optimizing all parameters in the perturbed model, including those of noise distributions, which is a reasonable proxy of the training without any noise.  Besides, noise injection plays a role in increasing the model capacity and regularization, and appropriate noise parameters can contribute to the performance and robustness.  Thus, we use ULR to calculate the gradient of noise distribution parameters with respect to classification loss and optimize them with gradient descent algorithms.  By viewing the training logs, we surprisingly find that the gradient estimation accuracy is also improved compared to that with the non-optimized noise, which indicates a good byproduct of the improved gradient estimation accuracy by noise optimization in ULR training. We appreciate the idea that noise parameters can be optimized for more accurate gradient estimation and hope this study can motivate the following works to derive new theoretical tools for this purpose.
>
> > **AQ3:** Yes, ULR-WL represents the hybrid perturbation manner in which we introduce noise to the weights of the first several layers and to the logits in the subsequent layers for gradient computation. In the case of ULR-WL training on VGG-8, we specifically perturb the weights in the first three layers and the logits in the remaining layers.
>
> > **AQ4:** In convolutional neural layers, each element within the same output channel shares a single bias scalar. Therefore, the whole bias term, denoted as $b$, consists of $c_\text{out}$ elements.

---

> > ### Comment · Reviewer_3HaG · 2023-11-20
> > **Additional questions**
> >
> > Thank you for your replies.
> >
> > I have another question regarding to your replies.
> > I agree that rearranging the gradient computation graph using ULR is an interesting direction.
> > But from the results in Figure 7, ULR achieves relatively low similarity compared to BP, especially in the middle layers, such as Conv-3 in the figure, even when the hybrid approach is used.
> > I think it is usually the middle layers that are applied ULR for the purpose of gradient computation graph rearrangement.
> > Do you have any comments on why the results in Section 4.2 achieve even higher accuracy than the full BP training, given the low gradient correlation observed in Figure 7?
> >
> > Regarding $b$ on page 4, I meant that $b^O$ is used in the equation, while $b$ is used in the paragraph below.
> > If these two represent different parameters, their relationship should be better explained.

---

> ### Author Response · Authors · 2023-11-21
>
> **AQ1:** This is largely because ULR does not suffer from the gradient vanishing/exploding issues that can occur with BP in deep forward propagations. Since gradients are calculated iteratively and multiplied by the derivative of the activation function at each layer/step in BP/BPTT, the gradient signal can decrease exponentially throughout the propagation when the derivatives of the activation function are close to zero. This can result in a stagnation of model performance growth [1,2]. However, in the BPTT rearrangement, ULR divides the gradient computation recursion into several shallower ones, mitigating this impact of the activation function. And in pure ULR, the gradient computation does not involve any differentiation of the activation function nor recursive calculations throughout the computation graph, thereby avoiding gradient-related issues.
>
> Moreover, the curves plotted in Fig. 7 represent the empirical means of cosine similarity, primarily reflecting the variance of our gradient estimation. ULR still yields an unbiased estimation of the true gradient. We observe that the depth of layers is not the primary factor affecting the variance. Rather, the results of pure ULR in Fig. 7 are due to both perturbation strategies using high dimensional noises in the middle layers. But the noise dimensionality can be reduced in the hybrid application with BP or further parallel of ULR. Additionally, the estimation variance does not necessarily suggest a failure of gradient descent, as the noise from an unbiased estimator, denoted by $\delta_k = g(\theta_k)- \nabla_{\theta}\mathbb E[\mathcal L(\theta_k)]$, where $\mathbb E[\delta_k] = 0$, can be averaged out throughout the gradient descent recursion [3]. Appropriate noise injection can even play a role in regularization, thus increasing the performance on the test dataset [4,5].
>
> [1] Pascanu, Razvan, Tomas Mikolov, and Yoshua Bengio. "On the difficulty of training recurrent neural networks." International conference on machine learning. PMLR, 2013.
> [2] Vorontsov, Eugene, et al. "On orthogonality and learning recurrent networks with long term dependencies." International Conference on Machine Learning. PMLR, 2017.
> [3] Harold, J., G. Kushner, and George Yin. "Stochastic approximation and recursive algorithm and applications." Application of Mathematics 35.10 (1997).
> [4] Zhou, Mo, et al. "Toward understanding the importance of noise in training neural networks." International Conference on Machine Learning. PMLR, 2019.
> [5] Smith, Samuel, Erich Elsen, and Soham De. "On the generalization benefit of noise in stochastic gradient descent." International Conference on Machine Learning. PMLR, 2020.
>
> **AQ2:** Thank you for your suggestion. We have specified the definition as "$b = (b^o)\in\mathbb R^{c_\text{out}}$" to clarify that $b^o$ is the $o$-th elememt of $b$ in the revision.

---

> > ### Comment · Reviewer_3HaG · 2023-11-21
> > **Thank you for your clarification**
> >
> > Thank you for your clarification.
> >
> > I understand why the results in Section 4.2 are improved by ULR, which may be due to the resolution of the vanishing/exploding gradient or the regularization effect of the gradient noise induced by ULR.
> >
> > I misunderstood that the gradient estimation of ULR can be biased compared to BP, for example from Figure 7.
> > Your comments above on the evaluation of cosine similarity are better noted in the manuscript.
> >
> > I still feel that the title "one forward is enough" is misleading, even though the authors claim that it means "one forward of a minibatch" including multiple $z^l$.
> >
> > I would like to keep my score.

---

> ### Author Response · Authors · 2023-11-21
>
> **A:** Thank you for your suggestion. We have modified the explanation of the cosine similarity in the manuscript. We also seek the reviewer's advice on whether "Only Forward is Necessary for Neural Network Training via the Likelihood Ratio Method" would be a more appropriate claim/title for this work.
>
> The current title has two intentions. Firstly, it aims to underscore the distinction between ULR and other algorithms like FF and SPSA, which require at least two forward passes per iteration. Although utilizing repeated evaluations is desirable for enhanced training quality, a single forward pass is theoretically acceptable for ULR when the numerical overflow issue is addressed with care, e.g., by parameter clipping. Secondly, the title emphasizes that ULR eliminates the need for recursive backward calculations, in contrast to forward computations.

---

> > ### Comment · Reviewer_3HaG · 2023-11-22
> > **Thank you for your modifications**
> >
> > Thank you for your modifications.
> >
> > If we don't limit ourselves to perturbation-based methods, there are several approaches that require only one forward computation, such as those using dual numbers as implemented in pytorch.
> > In this sense, I think that the title "Only Forward is Necessary" also does not accurately characterize the contribution of this paper, although I have no clear alternative.

---

> > > ### Author Response · Authors · 2023-11-22
> > >
> > > **A:** Thank you for your kind reminder. The uniqueness of ULR we would like to highlight lies in its **simultaneous** possession of three advantages. First, ULR does not require backward computation, thereby avoiding some numerical and efficiency issues associated with the deep recursion in BP. Second, ULR breaks the reliance on the chain rule of differentiation, allowing for the exploration of more diverse network structures and manners of gradient calculation. This is in contrast to forward-mode auto-differentiation and the dual number method, which require full knowledge of computation details in forward evaluations and necessitate the calculations to be differentiable [1]. Third, ULR does not give up the original optimization problem in ML or impose additional requirements on the model or optimizer. This enables ULR to be integrated with most existing ML achievements, which is not the case for many non-perturbation-based methods, such as FA, the HSIC bottleneck, and NTK compared in our paper.
> > >
> > > [1] Baydin, Atilim Gunes, et al. "Automatic differentiation in machine learning: a survey." Journal of Machine Learning Research 18 (2018): 1-43.

---

> > > > ### Comment · Reviewer_3HaG · 2023-11-23
> > > > **Thank you for your comments**
> > > >
> > > > I agree to the point that ULR has an advantage over existing forward mode auto-diff using dual numbers in the sense that ULR does not require full knowledge of the computational details of other layers.
> > > > And such a property seems useful for applications like rearranging the gradient computation graph.
> > > > I would like the author(s) to include a discussion of these points in the final manuscript.
> > > >
> > > > I have raised my score from 5 to 6.
> > > > Thank you for your responses.

---

> > > > > ### Author Response · Authors · 2023-11-23
> > > > >
> > > > > Thank you for your insightful suggestions and comprehensive discussions. We have modified our title/claim accordingly and included the discussion in the latest revision. The point mentioned by the reviewer opens up further possibilities for ML, which could be our next direction in the future.

---

### Official Review · Reviewer_s8CG · 2023-10-30

**Soundness:** 3 good
**Presentation:** 3 good
**Contribution:** 3 good
**Rating:** 6
**Confidence:** 4

**Summary:**

This work proposes the Unified Likelihood Ratio (ULR) a method that allows for more efficient neural network optimization. ULR is based on the old idea of zero-order optimization with likelihood ratio, reinterpreted in way that allows for optimising neural networks while avoiding back propagation, either entirely or some parts of it. By adding noise to either the activations or the weights, with the likelihood ratio method one only needs the loss (which can even be non-differentiable) evaluated at the noisy inputs in order to obtain a gradient for the terms that contributed to the noisy input (instead of the gradient of the loss itself with respect to the noisy input). The authors discuss how this idea generalises to various architectures, while also discussing three specific techniques that reduce the variance of the gradient estimates, which is very important for the practical success of a zero-order method. The authors evaluate ULR on several tasks and architectures while also performing an ablation study on their proposed variance reduction techniques and the impact of the perturbations on various points in the network.

**Strengths:**

- Simple method that works well in practice and tackles the practically relevant problem of efficient neural network optimization, especially given the recent successes of large models. The framework is quite general and the computation graph rearrangement via ULR is especially interesting.
- The experiments are quite diverse and informative.
- The paper is generally well-written.
- The variance reduction techniques highlighted, although quite standard, seem to help quite a bit in practice

**Weaknesses:**

- The work has a misleading claim; the authors argue that one forward pass is enough however in practice multiple forward passes are used in order to estimate the gradient (to reduce variance / antithetic variables). I would suggest that the authors rephrase the claim appropriately.
- The extension of LR to various network architectures is straightforward and not particularly novel.
- As far as I understand, the variance reduction techniques discussed at 3.3 seem to not be applied on the baselines, e.g., ES, so it is unclear where to attribute the gap in performance
- The architectures and settings are a bit toy; it would be interesting to see how ULR performs on much larger architectures (e.g., ResNet50 and Imagenet or GPT-2 on a reasonably sized text dataset)

**Questions:**

I find the overall paper interesting, especially the computation graph rearrangement part. Having said that, ULR seems to also be a combination of existing ideas and techniques, so it is not entirely novel. However, given the relatively good results, I am leaning towards acceptance. My questions and suggestions are the following

- Before section 3.2, the authors discuss about adaptive perturbations by optimising the noise, however, as far as I understand, this is not used anywhere in the paper. How is the performance with adaptive noise?
- On the RNN side, it seems that the authors add noise on each step, which can compound over time. Why is this strategy preferred compared to, e.g., just adding noise to the weights and evaluating the entire sequence on these noised weights (e.g., as done for convolutional layers).
- I would suggest the authors to incorporate the variance reduction techniques to ES as well, in order to better highlight what the delta with ULR is.
- At Figure 6, it is unclear what are the BP and ULR metrics.
- At Figure 7 what is the reference BP gradient compared against? The true gradient of the noised loss (i.e., the one that includes the noise from ULR) or the gradient of loss without any noise (which is the one we care about)?
- At the discussion of Figure 5 and Demo 1, the authors argue that the ULR can skip the calculation in late layers, however, this is not entirely true as ULR still needs the forward pass through those layers to compute the loss value.
- Finally, the accuracy of the base BP model on CIFAR 10 is kind of small; do the authors use data augmentations?

---

> ### Author Response · Authors · 2023-11-19
>
> ### Responses to Weaknesses
>
> > **AW1:** Thank you for your suggestion. We wonder whether the reviewer thinks "Only Forward is Necessary for Neural Network Training via the Likelihood Ratio Method" is an appropriate claim/title for this paper. The initial intent of the current title is to highlight the distinction between our ULR algorithm and others like FF and SPSA, which require at least two forward passes, as well as the absence of a need for recursive backward calculations in contrast to forward ones. Although using multiple evaluations is preferable for enhanced training quality, one forward pass is theoretically acceptable. Furthermore, multiple evaluations can be packed into a single batch to be processed in one forward pass.
>
> > **AW2:** The purpose of these extensions is to provide readers with specific examples, opening up ideas for the use of ULR. We offer corresponding solutions for issues related to spatio-temporal parameter sharing, unstructuredness, and discontinuity, which cover almost all possible network structures in mainstream deep learning, demonstrating the flexibility of ULR training. This part also inspires our subsequent discussions on computation graph rearrangement and the relationships between different works/paradigms.
>
> > **AW3:** Please see the response in AQ3.
>
> > **AW4:** ULR offers better scalability than previous methods for training larger models, but using it exclusively on large neural networks may not be that cost-effective. Replacing deterministic calculations with stochastic ones may require repeated loss evaluations. Thus, in our paper, we conduct experiments with pure ULR as well as other baselines on relatively shallow neural networks to verify the effectiveness of ULR training. Since ULR is compatible with BP, a viable application prospect in large neural networks is to use ULR to skip the backward computation in non-updating parts of the network or to transform iterative computations into parallel ones, as demonstrated in Section 4.2.

---

> ### Author Response · Authors · 2023-11-19
>
> ### Responses to Questions 1-3
>
> > **AQ1:** In ULR experiments, we optimize parameters in noise distributions during training for enhanced performance. ULR supports the estimation of gradients for all parameters within the neural network. We calculate the gradients of loss values with respect to the noise generation parameters and apply gradient descent algorithms, such as SGD or Adam, to update these parameters. The estimated gradient formulas for noise distribution parameters are presented in Section 3.1 and Appendices A.1-5. With the adaptive perturbation technique, the performance of neural networks can be further improved.
> > We conduct ablation studies on adaptive perturbation. Following the same setting in our paper, we respectively use ULR with and without adaptive perturbation to train ResNet-5 and VGG-8 on the CIFAR-10 dataset, and RNN, GRU, and LSTM on the Ag-News dataset. The results are reported as follows.
> >
> > Tab. R1 Classification accuracies (\%) of ResNet-5 and VGG-8 on the CIFAR-10 dataset. We respectively use ULR with ($\checkmark$) and without ($\times$) adaptive perturbation in neural network training.
> >
> >| Models | Adaptive Perturbation | Accuracy |
> >| -------- | -------- | -------- |
> >| ResNet-5     | $\times$     |  61.3    |
> >| ResNet-5     | $\checkmark$     |  64.8    |
> >| VGG-8     | $\times$     |  75.9    |
> >|VGG-8     | $\checkmark$     |  77.3    |
> >
> > Tab. R2 Classification accuracies (\%) of RNN, GRU, LSTM on the Ag-News dataset. We respectively use ULR with ($\checkmark$) and without ($\times$) adaptive perturbation in neural network training.
> >
> >| Models | Adaptive Perturbation | Accuracy |
> >| -------- | -------- | -------- |
> >| RNN     | $\times$     |  69.2    |
> >| RNN     | $\checkmark$     |  70.9    |
> >| GRU     | $\times$     |  86.6    |
> >| GRU     | $\checkmark$     |  88.9    |
> >| LSTM     | $\times$     |  87.8    |
> >| LSTM     | $\checkmark$     |  89.4    |
> >
> >As reported in Tab. R1 and R2, we observe an improvement in neural network training with ULR by adopting adaptive perturbation. This demonstrates the effectiveness of optimizing the noise distribution in boosting the accuracy of gradient estimation for ULR.
>
> > **AQ2:** Thank you for your insightful suggestion. ULR supports various noise injecting strategies, which hinge on how we define the module $\varphi^l(\cdot;\theta^l)$ in Theorem 1. Injecting noise into the weights can be interpreted as perturbing the logit output of an identity mapping, with the weights as its input. And we agree that there may be better strategies to apply our ULR framework. As mentioned in the CNN part, the variance of the gradient estimation grows as the total dimensionality of injected noises increases. Therefore, the choice between the step-wise and weight perturbation largely depends on $T$ and $d_x$. Furthermore,  if the impact of each step on the final loss is additive, then we can obtain a more refined result for RNNs with step-wise perturbation as shown in Appendix A.7. For scenarios where the effect of a single forward step is transient, truncation tricks can be employed to further reduce the impact of $T$, akin to the introduction of a discount factor in reinforcement learning. Consequently, when $T$ is manageable, and the loss is decomposable, the noise injection strategy outlined in our paper is preferred. In contrast, we may consider the weight perturbation mentioned above.
>
> > **AQ3:** We apply our proposed methods for variance reduction, including layer-wise noise injection, antithetic variables, and Quasi-Monte Carlo, to the ES algorithm. We use ES to train ResNet-5 and VGG-8 on the CIFAR-10 dataset. For comparison, we also present the results of ULR in our paper. The results are as follows.
> >
> >Tab. R3 Classification accuracies (\%) of ResNet-5 and VGG-8 on the CIFAR-10 dataset using the ES and ULR, respectively.
> >
> >| Model | ResNet-5 | VGG-8 |
> >| -------- | -------- | -------- |
> >| ES  without tricks   | 35.1     | 65.8     |
> >| ES  with tricks   | 46.9     | 67.2     |
> >| ULR     | 64.8     | 77.3     |
> >
> > As shown in Tab. R3, with the proposed tricks, there is a significant improvement in ES for training neural networks, indicating the effectiveness of our tricks in variance reduction. However, even with the integration of these tricks, ES still struggles to optimize neural networks effectively, demonstrating a lower capacity in gradient estimation compared to ULR. This argument is also supported by our ablation study in Fig. 7(b) of Section 4.3. When perturbing only the weights for gradient estimation, we achieve considerable accuracy in only the first one or two layers but fail to calculate the rest accurately. This further highlights the superiority of ULR over ES in gradient estimation, as shown in Fig. 7\(c\).

---

> ### Author Response · Authors · 2023-11-19
>
> ### Responses to Questions 4-7
>
> > **AQ4:** In Fig. 6, we present a performance evaluation of BPTT rearrangement. The evaluation metrics include the **classification accuracy**  of trained RNNs on the Ag-News dataset and the **training duration** per epoch, where the classification accuracy and training duration are represented by the curves and bars, respectively. The x-axis represents the depth of the gradient computation graph used in BPTT boosted by ULR, while 'BPTT' on the right side indicates no cuts on the graph. The number '5' indicates that we cut the original graph into multiple subgraphs with a maximum depth of 5 and use ULR to launch the gradient computation at the leaf node in each subgraph, then apply BPTT to pass information throughout the rest. It can be observed that with BPTT boosted by ULR, the training efficiency of RNNs is improved, where the classification accuracy remains high, but the training duration is reduced.
>
> > **AQ5:** In Fig.7, the reference BP gradient is compared with the estimated gradient by ULR. In ULR, the estimated gradient is computed by injecting noise into the loss.  At the same time, the BP gradient is computed on the vanilla network, where no noise is injected into neural activities, thus excluding the effect of perturbation. We use the cosine similarities to evaluate the gradient estimation accuracy.
>
> > **AQ6:** We agree that the application of ULR cannot avoid the forward pass on non-trainable/frozen layers. The advantage of ULR is to skip the gradient computation of these layers during the backward pass, thus bringing an efficiency improvement as presented in Tab. 1. As pointed out by previous studies [1,2], one backward pass in a neural network theoretically takes about twice as long as the forward pass. Computation hardware, such as GPUs and TPUs, supports the forward pass far better than the backward pass [3]. Thus, by disabling part of the backward pass in gradient computation, ULR can achieve efficiency improvement in practice in our experiments as well as in theoretical analysis. Moreover, ULR can take advantage of the importance sampling technique in the future to update parameters several times with only one forward pass, which further reduces unnecessary calculations.
> >
> > [1] Lin, Zhouhan, et al. "Neural networks with few multiplications." arXiv preprint arXiv:1510.03009 (2015).
> > [2] Narayanan, Deepak, et al. "PipeDream: Generalized pipeline parallelism for DNN training." Proceedings of the 27th ACM Symposium on Operating Systems Principles. 2019.
> > [3] Liu, Xiao-Yang, et al. "High-Performance Tensor Learning Primitives Using GPU Tensor Cores." IEEE Transactions on Computers (2022).
>
> > **AQ7:** We only use one augmentation strategy, namely the random horizontal flip, in our model training on the CIFAR-10 dataset. All the parameters have been fine-tuned for the best performance, and the models are trained sufficiently to converge. However, due to the limited size of the neural networks, i.e., ResNet-5 and VGG-8, they do not achieve high accuracy on the CIFAR-10 dataset, even with BP for gradient computation. Similar results can be found in a previous study [1], which used BP to train a shallow ResNet and achieved around 60% accuracy on the CIFAR-10 dataset. We also compare this work in our appendix of Section B.6.
> >
> > [1] Ma, Wan-Duo Kurt, J. P. Lewis, and W. Bastiaan Kleijn. "The HSIC bottleneck: Deep learning without back-propagation." AAAI, 2020.

---

> ### Author Response · Authors · 2023-11-22
>
> Dear Reviewer s8CG,
>          We have submitted our response to your questions. We sincerely appreciate your valuable feedback on improving the quality of our paper.
>     Are there any additional questions or concerns we can answer? Thanks for your reply!
>
> Yours,
> Authors

---

> > ### Comment · Reviewer_s8CG · 2023-11-22
> > **Response to rebuttal**
> >
> > I would like to thank the authors for their extensive responses to my questions. These have resolved most of my concerns and the replacement title, although not ideal as reviewer 3HaG also points out, it is a better fit compared to the existing one. I thus encourage the authors to update the current manuscript accordingly. I am retaining my score for now but I will update it after the authors upload their revision.

---

> > > ### Author Response · Authors · 2023-11-23
> > >
> > > Thank you for your valuable suggestions. We have come up with a new solution for our title/claim and included the additional experiments above in the latest revision.

---

### Official Review · Reviewer_hjbZ · 2023-10-31

**Soundness:** 4 excellent
**Presentation:** 4 excellent
**Contribution:** 3 good
**Rating:** 8
**Confidence:** 3

**Summary:**

This paper proposes a unified likelihood ratio (ULR) method to avoid the recursive computation of backpropagation. By imposing noise to input data in forward propagation, ULR can approximate the gradient independently for each layer in backpropagation. The authors show how ULR can be applied to architectures such as DNNs, CNNs, RNNs, GNNs, and SNNs, and also introduce techniques to reduce the variance of ULR. The proposed method not only performs comparably to backpropagation, but is also more efficient than backpropagation in some highly recursive computationally demanding tasks.

**Strengths:**

This paper proposes a unified likelihood ratio (ULR) method to avoid the recursive computation of backpropagation. By imposing noise to input data in forward propagation, ULR can approximate the gradient independently for each layer in backpropagation. The authors show how ULR can be applied to architectures such as DNNs, CNNs, RNNs, GNNs, and SNNs, and also introduce techniques to reduce the variance of ULR. The proposed method not only performs comparably to backpropagation, but is also more efficient than backpropagation in some highly recursive computationally demanding tasks.

**Weaknesses:**

Due to layer-wise noise injection, which is one of the proposed variance reduction techniques, ULR has a forward propagation cost that is proportional to the number of layers. When applied to deep networks with many layers (ResNet-152, BERT-large, etc.), it may have a high computational cost compared to backpropagation. The NNs used in the experiments have a small number of layers (ResNet-5, VGG-8), which does not address this concern.

**Questions:**

Why did you plot the experiments in Figures 7 and 11 separately for each layer? What observations can we make there?

---

> ### Author Response · Authors · 2023-11-19
>
> ### Responses to Weaknesses
>
> > **A:** We agree with the reviewer that consumption will increase for ULR to produce accurate gradient estimations as the scale of neural networks grows. There remain three promising directions to alleviate this issue. Firstly, since ULR is compatible with BP, using BP in modules with good continuity and ULR in black boxes, non-updating, or discontinuous parts can be a more economical option than using either BP or ULR solely, as we demonstrated in Sections 3.4 and 4.2. The perturbation occurs only in the parts where ULR is applied, thus making the consumption acceptable. Secondly, the importance sampling technique has the potential to be integrated into ULR with care, which allows multiple updates with one forward evaluation and further reduces the computational effort. Thirdly, we might consider relaxing the requirements for extremely accurate gradient estimations and exploring robust optimizers that match ULR, where the impact of the estimation variance can be averaged out during iterations.
>
> ### Responses to Questions
>
> > **A:** We intend to study the role of the proposed tricks presented in this paper through ablation experiments. The cosine similarity between gradients computed by ULR and BP serves as a proxy indicator for network training quality. We observe that the accuracy of gradient estimation varies across different neural layers and exhibits a 'buckets effect' during training. Hence, it is necessary to plot each layer separately to analyze the effect of these proposed tricks. Firstly, as shown in the first row of Fig. 7 and 11, perturbing the logits results in improved figures when the input dimensionality is small and the weight dimensionality is large, aligning with the discussion in Section 3.2. Secondly, the middle row of Fig. 11 indicates that the optimal choice of initial noise magnitude depends on the structure and order of the layers. Thirdly, we visualize the effect of the proposed variance reduction methods in the last row of Fig. 7 and 11. Their absence consistently negatively impacts all layers.

---

> ### Author Response · Authors · 2023-11-22
>
> Dear Reviewer hjbZ,
>          We have submitted our response to your questions. We sincerely appreciate your valuable feedback on improving the quality of our paper.
>     Are there any additional questions or concerns we can answer? Thanks for your reply!
>
> Yours,
> Authors

---

### Official Review · Reviewer_HFF5 · 2023-11-05

**Soundness:** 4 excellent
**Presentation:** 3 good
**Contribution:** 3 good
**Rating:** 8
**Confidence:** 3

**Summary:**

This paper presents a unified framework for training neural networks with likelihood ratio estimation methods.
Likelihood ratio estimation has the benefit of avoiding backpropagation for gradient computation, thus leveraging the backward lock.
Such optimization methods are thus amenable to greater parallelization, as gradient computation for different modules can be performed independently of each other given the final loss evaluation.
The authors combine two variance reduction techniques to make the technique practical.
The proposed method is evaluated on a diverse set of experiments covering various deep learning applications.
The ablation studies cover most of the critical components of the proposed method.
The paper may lack some discussion about the practical limitations but is overall well-written, with some sparse exceptions, and the research is well-conducted.

**Strengths:**

1) The paper is dense, with all the details needed to implement the proposed method. In particular, gradient estimators for the parameters of ubiquitous neural architecture components are provided.
2) The proposed framework offers large flexibility with respect to the practical implementation, e.g. for model parallelism.
3) The proposed set of experiments covers a diverse range of applications in deep learning. The ablation study in the main body covers important aspects of the proposed method, such as the type of perturbation, as well as the practical relevance of the variance reduction techniques employed. Many additional experiments are provided in the appendix.

**Weaknesses:**

1) I don't understand theorem 1. It seems that the function $g^l$ is implicitly defined in the proof. But I don't understand if this translates into a condition on $f^l$ in the hypothesis of the theorem, and if so, why the condition does not directly involve $f^l$. The function $g$ is not a component of the gradient estimator, so I don't understand why it is present in the hypothesis. Also, the variable $\xi$ seems to represent $z^l$ (or ($z^{l+1}$) and I'm not sure what this condition implies.
2) Does the star in equation (3) represent a convolution or a term-by-term product?
3) The set of experiments chosen are a bit simple. While it is, for example, promising to produce good classification accuracy for MNIST and CIFAR10, the question of scaling these methods to larger models or more complex datasets is not being discussed.
4) [Minor comments] "Since these methods change the classical training paradigm or even give up the
gradient information. Many algorithmic or hardware technologies developed based on BP by predecessors cannot be fully leveraged, making such approaches computationally unfriendly." this might have been a single sentence.

**Questions:**

1) I may have missed it, but it's not clear from the text whether all parameters should be perturbed independently. It seems that it would help to have a better correlation between the noises and the loss, but this does not seem to be the case in the experiments.
Is there a middle ground between only perturbing the logits of the last layer, all layers, or all neurons independently?
2) Could the authors provide some hints on the current practical limit of the method with respect to model size and dataset?
For example, does this method scale for a resnet18 or resnet50 on CIFAR10 or ImageNet?
3) [Minor question] "LR has almost no constraints on the model, including differentiability or traceability, enabling explorations of network architectures." --> Could you explain how it enables explorations of network architectures?

---

> ### Author Response · Authors · 2023-11-19
>
> ### Responses to Weaknesses
>
> > **AW1:** Thank you for your suggestion. We will modify the assumptions of Theorem 1 in our revision. In our derivation, the expectation of the loss is expanded as follows: $$\mathbb E_{z^0,\cdots, z^{L-1}}[\mathcal L(x^L)] = \mathbb E_{x^l}[\mathbb E_{x^{l+1}}[\mathbb E_{z^{l+1},\cdots, z^{L-1}}[\mathcal L(x^L)|x^{l+1},x^l]|x^l]],$$ where we have reformulated the second expectation as an integral in the paper. Thus, the variable $\xi$ represents $z^l$, and $g^l(\cdot)$ denotes the conditional density of $x^{l+1}$ given $x^l$. By applying the law of iterated expectation, ULR pushes the parameter $\theta^l$ from the loss evaluation $\mathcal{L}(x^L)$ into the conditional density $g^l(\cdot)$, restricting the differentiation within $g^l(\cdot)$ as shown Fig. 2. The conditions involve not only $f^l(\cdot)$ but also $\varphi^l(\cdot;\cdot)$, and we intend to encapsulate them using $g^l(\cdot)$ for readability. Moreover, since $\varphi^l(\cdot;\cdot)$ can be defined flexibly, excluding those non-differentiable but non-parametric items, whether the assumptions of Theorem 1 are met primarily hinges on the smoothness of $f^l(\cdot)$.
>
> > **AW2:** Yes. The star in Eq. (3) represents the convolution of $x^i$ with kernel $\varepsilon^o$. That is the reason why ULR for CNNs can be implemented with existing high-performance convolution algorithms in standard machine learning toolkits, e.g., `torch.nn.functional.conv2d` in PyTorch. More details can be found in lines 58--71 of cnn/module.py, which is provided in the supplementary.
>
> > **AW3:** Please see the response in AQ2.
>
> > **AW4:** Thanks. We fix this typo in the revision as follows: "Since these methods change the classical training paradigm or even give up the gradient information, many algorithmic or hardware technologies developed based on BP by predecessors cannot be fully leveraged, making such approaches computationally unfriendly."
>
> ### Responses to Questions
>
> > **AQ1:** There are two extreme perturbing strategies for gradient estimation. One strategy is to perturb different neurons independently and precisely compute the gradients of parameters in each neuron. However, it is computationally intensive and leads to unbearable training efficiency. The other perturbs all neurons of the whole neural network. While it requires less computation for each estimation, it results in lower gradient estimation accuracy and, consequently, diminished neural network performance. In our paper, we further propose a layer-wise perturbation strategy, perturbing the neurons in one layer at a time. This design strikes a balance between gradient estimation accuracy and computation efficiency. ULR allows flexible perturbation across different parts of the neural network, depending on the consideration of the location of optimizable parameters, the training efficiency, and the gradient estimation accuracy.
>
> >**AQ2:** The current practical limitation of ULR is the large number of data copies required for accurate gradient estimation when training large models on large datasets, which imposes an additional burden on device memory. Training ResNet-50 on ImageNet using pure ULR may be uneconomical due to the expensive memory footprint.
> >
> >However, as presented in Sections 3.4 and 4.2, ULR is compatible with BP and can be used to segment the gradient computation graph into multiple subgraphs. In each subgraph, gradient computations from leaf nodes are initiated independently and in parallel using ULR, while BP transmits information throughout the rest. This design doesn’t require as many copies for accurate gradient computation and can make a trade-off between parallelism and repeated forward evaluation, which may enable us to train large models on large datasets with promoted efficiency. Thus, we hope our study can motivate future work on 1) proposing novel techniques to reduce the gradient estimation variance in ULR in a computation- and memory-friendly manner; 2) designing an efficient training framework by rearranging the gradient computational graph with ULR.
>
>
> >**AQ3:** ULR offers the flexibility to wrap any structure as a unit for independent gradient computation, which only requires the unit to be differentiable. This feature enables the selection of various structures for the untrainable or non-parametric components within neural networks. The direct impact is that we do not have assumptions of activation functions. In Section 4.1, we effectively train SNNs with ULR, where the non-differentiable spiking activation function hinders the application of vanilla BP. From a broader perspective, ULR enables the integration of black box elements like simulators, pre-trained large models, and commercial optimizers into neural networks as static modules. This significantly facilitates the fusion of domain-specific knowledge with neural network architectures. In Section 4.2, the first demo for domain adaptation provides an example of this purpose.

---

> > ### Comment · Reviewer_HFF5 · 2023-11-22
> > **Response to authors rebuttal**
> >
> > I kindly thank the reviewers for their detailed response.
> >
> > Response to weaknesses:
> > > AW1: Thank you for clarifying the assumptions and notations of theorem 1.
> >
> > > AW2: I could not check the math again, but I was suspicious of the convolutional operator. My intuition was that it would have been a term-by-term product so thanks for the clarification. I acknowledge that such an estimator can take advantage of optimized implementation of convolutional operators which are readily available in most deep learning library.
> >
> > Response to questions:
> > > AQ1 & AQ2: Thanks to the authors for clarifying the efficiency vs accuracy tradeoffs of the proposed method. Given this explanation, it would be nice to see a better ablation on this matter to evaluate the parallelization potential of the given method. For example, a ResNet5 could be split at each layer or at each residual blocks. Splitting at each layer thus enables greater parallelization potential at the expense of less accurate gradient estimate. However, the parallelization potential may enable more forward pass to happen within the amount of time needed for a full forward propagation of a mini-batch. Thus, a finer ablation on this specific tradeoff would be much valuable to evaluate the scalability of the proposed method. That being said, this could reasonably be delayed for further investigation given the amount of ablation already reported in the presented work. I also note that optimally rearranging the gradient computational graph with respect to a given hardware configuration appears to be a non-trivial task rarely addressed in the machine learning literature.
> >
> > > AQ3: Thanks to the authors for clarifying my understanding on the neural architecture exploration potential of the proposed method. I agree that the ability to integrate non-differentiable components in the computational graph provides an advantage over standard backpropagation.
> >
> > I will raise my score accordingly because, even though I still have concerns about the practical relevance and scalability of the proposed method large datasets and complex models, I find that the experiments appropriately covers important aspects such as the impact of the perturbation strategy or the variance reduction techniques (given the answer to question AQ1 of reviewer s8CG). I think this paper may be a good starting point for further practical investigations if the ULR technique happens to be competitive with backpropagation when integrating engineering tricks and appropriate hyperparameter tuning in a distributed setting for large models.

---

> > > ### Author Response · Authors · 2023-11-23
> > >
> > > Thank you for your extensive inspiration. Focusing on the original mathematical problem of learning, ULR is highly compatible with existing engineering tricks. We will keep improving the practice of ULR for enhanced scalability, facilitating more possibilities in the future.

---

> ### Author Response · Authors · 2023-11-22
>
> Dear Reviewer HFF5,
>          We have submitted our response to your questions and revised our paper as you suggested. We sincerely appreciate your valuable feedback on improving the quality of our paper.
>     Are there any additional questions or concerns we can answer? Thanks for your reply!
>
> Yours,
> Authors

---

### Comment · Area_Chair_YGeV · 2023-11-22

Dear all,

The author-reviewer discussion period is about to end.

@authors: If not done already, please respond to the comments or questions reviewers may further have. Remain short and to the point.

@reviewers: Please read the author's responses and ask any further questions you may have. To facilitate the decision by the end of the process, please also acknowledge that you have read the responses and indicate whether you want to update your evaluation.

You can update your evaluation positively (if you are satisfied with the responses) or negatively (if you are not satisfied with the responses or share other reviewers' concerns). Please note that major changes are a reason for rejection.

You can also keep your evaluation unchanged. In this case, please indicate that you have read the responses, that you do not have any further comments and that you keep your evaluation unchanged.

Best regards,
The AC

---

### Meta-Review · Area_Chair_YGeV · 2023-12-08

**Metareview:**

The reviewers unanimously recommend acceptance (8-8-6-6). The paper proposes a unified likelihood ratio method for gradient estimation, effectively providing an alternative to backpropagation. The reviewers note the dense contributions of the paper, ranging from theoretical results to extensive empirical evaluation. The author-reviewer discussion has been constructive and has led to a number of improvements to the paper.

**Justification For Why Not Higher Score:**

Despite the good scores, the reviewers have raised a number of minor concerns and reservations on the scalability, relevance and novelty.

**Justification For Why Not Lower Score:**

All reviewers recommend acceptance.

---

### Decision · Program_Chairs · 2024-01-16

Accept (poster)